# Murine alveolar macrophages rapidly accumulate intranasally administered SARS-CoV-2 Spike protein leading to neutrophil recruitment and damage

Chung Park[1], Il-Young Hwang[1], Serena Li-Sue Yan[1], Sinmanus Vimonpatranon[2,3], Danlan Wei[2], Don Van Ryk[2], Alexandre Girard[2], Claudia Cicala[2], James Arthos[2], John H Kehrl[1]*

[1]B-Cell Molecular Immunology Section, Laboratory of Immunoregulation, National Institute of Allergy and Infectious Diseases, National Institutes of Health, Bethesda, United States; [2]Immunopathogenesis Section, Laboratory of Immunoregulation, National Institute of Allergy and Infectious Diseases, Bethesda, United States; [3]Department of Retrovirology, Armed Forces Research Institute of Medical Sciences – United States Component, Bangkok, Thailand

*For correspondence:
jkehrl@niaid.nih.gov

Competing interest: The authors declare that no competing interests exist.

**Abstract** The trimeric SARS-CoV-2 Spike protein mediates viral attachment facilitating cell entry. Most COVID-19 vaccines direct mammalian cells to express the Spike protein or deliver it directly via inoculation to engender a protective immune response. The trafficking and cellular tropism of the Spike protein in vivo and its impact on immune cells remains incompletely elucidated. In this study, we inoculated mice intranasally, intravenously, and subcutaneously with fluorescently labeled recombinant SARS-CoV-2 Spike protein. Using flow cytometry and imaging techniques, we analyzed its localization, immune cell tropism, and acute functional impact. Intranasal administration led to rapid lung alveolar macrophage uptake, pulmonary vascular leakage, and neutrophil recruitment and damage. When injected near the inguinal lymph node medullary, but not subcapsular macrophages, captured the protein, while scrotal injection recruited and fragmented neutrophils. Widespread endothelial and liver Kupffer cell uptake followed intravenous administration. Human peripheral blood cells B cells, neutrophils, monocytes, and myeloid dendritic cells all efficiently bound Spike protein. Exposure to the Spike protein enhanced neutrophil NETosis and augmented human macrophage TNF-α (tumor necrosis factor-α) and IL-6 production. Human and murine immune cells employed C-type lectin receptors and Siglecs to help capture the Spike protein. This study highlights the potential toxicity of the SARS-CoV-2 Spike protein for mammalian cells and illustrates the central role for alveolar macrophage in pathogenic protein uptake.

## eLife assessment

This paper investigates the impact of intranasal instillation of SARS CoV2 spike protein in mouse models of lung inflammation. The authors conclude that the spike protein can interact with macrophages through carbohydrate recognition and can induce recruitment and NETosis of neutrophils, contributing to lung inflammation. They also use the cremaster muscle model to investigate effect of the spike proteins on neutrophil dynamics and death using intravital microscopy. Given that mucosal vaccines using SARS CoV2 spike variants could be envisioned as desirable, the observation that spike can induce lung/mucosal inflammation even without an adjuvant is **important**. Despite

limitations of some loose terminology and some weak controls, the key observations are **solid** and demand further attention given the importance of the antigen.

## Introduction

In 2019 a new coronavirus (SARS-CoV-2) was identified as the cause of an epidemic outbreak of an acute respiratory syndrome in Wuhan, China. SARS-CoV-2 used the same cell entry receptor—angiotensin converting enzyme II (ACE2)—as did SARS-CoV-1 (*Zhou et al., 2020*). The SARS-CoV-2 Spike protein mediates cell entry and is a single-pass transmembrane proteins that forms homotrimers. It has a large N-terminal ectodomain exposed to the exterior, a transmembrane helix, and a short C-terminal tail located within the virus. Each Spike monomer contains two regions termed S1 and S2. In the assembled trimer the S1 regions contain the receptor-binding domain while the S2 regions form a flexible stalk, which mediates membrane fusion between the viral envelope and the host cell membrane (*Fan et al., 2020*; *Ke et al., 2020*; *Wrapp et al., 2020*). The S1 region is divided into an N-terminal domain and a C-terminal domain, the latter interacts with the target cell ACE2 (*Li et al., 2005*; *Benton et al., 2020*; *Lan et al., 2020*; *Ou et al., 2020*; *Shang et al., 2020a*; *Shang et al., 2020b*; *Yan et al., 2020*). Like other coronavirus Spike proteins, SARS-CoV-2 Spike protein is heavily N-linked glycosylated (*Walls et al., 2016*; *Lenza et al., 2020*) and blocking N- and O-glycans dramatically reduced viral entry (*Yang et al., 2020*). The SARS-CoV-2 Spike proteins are activated by host cell proteases that cleave the protein at the S1-S2 boundary and subsequently at the S2' site (*Hoffmann et al., 2020a*, *Hoffmann et al., 2020b*, *Takeda, 2022*). Because Spike proteins are located on the surface of the virus, they are a major antigen targeted by the host immune system (*Errico et al., 2022*).

During natural infection host immune cells encounter Spike proteins via several different avenues. First, by direct contact with Spike protein bearing viral particles released from infected cells. Second, although the SARS-CoV-2 virions assemble in intracellular compartments of infected cells, unincorporated Spike proteins can reach the plasma membrane. Infected cells expressing Spike proteins may bind to cellular receptors present on resident or recruited immune cells. Third, extracellular vesicles (EVs) released by virally infected cells can contain Spike proteins. Mass spectrometry and nanoscale flow cytometry demonstrated SARS-CoV-2 Spike protein incorporation into EVs (*Troyer et al., 2021*). Fourth, Spike proteins are preprocessed during viral assembly utilizing the furin protease cleavage site between the S1 and S2 subunits (*Takeda, 2022*). When the Spike protein adopts a fusion conformation, the ACE2 receptor-binding domain separates from the membrane-bound S2 subunit. Soluble S1 subunits shed from infected cells or from the virions in vivo may bind to other cells via ACE2 or other binding partners. Intriguingly, 60% of the plasma samples from patients with post-acute sequelae of coronavirus disease 2019 had detectable levels of Spike protein using an ultrasensitive antigen capture assay (*Swank et al., 2022*). Prolonged exposure to Spike protein has been suggested to be responsible for Long-COVID syndrome (*Theoharides, 2022*). Humans also encounter SARS-CoV-2 Spike protein via vaccination (*Thanh Le et al., 2020*). Serious adverse events following vaccination are rare but include vaccine-induced immune thrombocytopenia and thrombosis; myocarditis and pericarditis; and a variety of autoimmune illnesses (*Lamprinou et al., 2023*). Due to waning effectiveness humans require repeated immunizations to maintain immunity and protection against potentially severe disease raising some concerns about the impact of repeated exposure to the SARS-CoV-2 Spike proteins.

To better understand the localization and trafficking of the SARS-CoV-2 Spike protein following administration and perhaps during natural infection, we prepared a recombinant SARS-CoV-2 Spike ectodomain stabilized in a prefusion conformation (*Hsieh et al., 2020*). This variant (S-2P) contained two consecutive proline substitutions in the S2 subunit. This double-proline substitution (SARS-CoV-2 S-2P) has allowed the rapid determination of high-resolution cryo-EM structures. Fluorescently labeled SARS-CoV-2 Spike ectodomain, a D614G variant (*Yurkovetskiy et al., 2020*; *Zhang et al., 2020*; *Volz et al., 2021*), a high mannose (*Yang et al., 2020*), and de-glycosylated version were injected into mice to characterize uptake and identify human mononuclear cells that bound these proteins, and to perform functional studies. In some instances, we used viral like particles (VLPs) expressing the full-length Spike protein. Our results identified the in vivo cellular tropism of the SARS-CoV-2 Spike proteins, delineated their human mononuclear cell targets, provided insights into their functional effects, thereby, helping afford a basis for understanding their impact on humans.

## Results

### Preparation of SARS-CoV-2 Spike proteins and VLPs expressing them

We purified a stabilized exodomain of the original SARS-CoV-2 Spike protein (*Hsieh et al., 2020*) using media conditioned by transfected CHO-S or HEK293F cells. Because of higher yields most experiments used CHO-S-derived protein. The strategy for producing the recombinant protein is shown (*Figure 1—figure supplement 1*). Based on its mobility on size exclusion chromatography, the Spike proteins spontaneously formed trimers. Kifunensine-treated cultures were used to generate a high mannose version, and PNGase F-treated protein generated an N-glycan deficient version, and they will be referred to as such. The PNGase F treatment resulted in two distinct chromatography peaks that exhibited slightly different mobilities on SDS-PAGE. As discussed below we predominately used the preparation from fractions 9–11. The D614G mutant of the original Wuhan SARS-CoV-2 Spike protein was also purified. Each of the recombinant proteins was subjected to two rounds of Triton X-114 extraction to remove any residual lipopolysaccharide (LPS) (*Aida and Pabst, 1990*). The N-termini of the recombinant proteins were labeled with Alexa Fluor 488. In some experiments we used VLPs that expressed the original Wuhan Spike protein. We produced the VLPs using HEK293T cells along with a plasmid encoding the human immunodeficiency GAG protein fused to green fluorescent protein (GFP) and a full-length SARS-CoV-2 Spike protein expression vector. Mice received the Spike protein preparations or the SARS-CoV-2 VLPs by intranasal instillation, while only the Spike proteins were injected intravenously or subcutaneously.

### Instillation of SARS-CoV-2 Spike proteins and Spike protein expressing VLPs affects the lung cellular composition and architecture

At various time points after Spike protein or VLP nasal instillation lungs from mice were processed for confocal microscopy. A Siglec-F antibody identified alveolar macrophages (AMs) while CD169 (Siglec-1) immunostaining delineated a subset of large airway-associated macrophages (*Hetzel et al., 2021*). Interstitial and inflammatory macrophages do not express Siglec-F. At 3 hr post instillation, low-magnification images of mouse lung sections revealed Spike protein (Trimer) uptake by Siglec-F-positive cells. Due to the uneven distribution of the instilled Spike protein, the Siglec-F-positive cells in the lower portion of the image lack signal (*Figure 1A*, *Figure 1—figure supplement 2A*). The Siglec-F-positive macrophages located near the bronchial epithelium and in the nearby alveoli initially accumulated the labeled protein (*Figure 1B*). Even at the 30 min time point AMs had already accumulated it (*Figure 1C*). Most of the macrophages that acquired the Spike proteins were Siglec-F positive, lacked CD169 (*Figure 1D*), but were wheat germ agglutinin positive, a mucin marker (*Figure 1E*). Repeating the experiment but substituting the Spike protein with VLPs expressing the SARS-CoV-2 Spike protein revealed a similar pattern of uptake by AMs, but with greater neutrophil recruitment and granulation compared to delta Env VLPs (*Figure 1F*, *Figure 1—figure supplement 2B and C*). Immunostaining with LYVE-1 and CD31 to identify the lymphatic and blood vessel endothelium, respectively (*Figure 1G*), or with Podoplanin, another lymphatic endothelial cell marker (*Figure 1H*, *Figure 1—figure supplement 2D*) showed the rapid accumulation of the Spike protein bearing VLPs in small and large lymphatic vessels. Within 3 hr of administration of either the VLPs (*Figure 1I*) or the Spike protein (data not shown), areas of alveolar collapse and lung damage were evident.

To assess the cellular response to Spike protein instillation and to delineate the cells that had acquired Spike protein in vivo, we collected mouse lungs 18 hr post exposure to different SARS-CoV-2 Spike protein preparations. Saline instillation served as a control. Lung fragments were digested, and then pushed through a 40 μm cell strainer. The cells collected were analyzed using flow cytometry following the established gating strategy (*Yu et al., 2016*). The results showed that the SARS-CoV-2 Spike protein increased the number of neutrophils, monocytes, and dendritic cells in the lung, but did not affect lymphocyte or eosinophil numbers (*Figure 2A*). Due to the uneven distribution of the instilled protein and because of their rapid turnover, the neutrophil numbers likely underestimate the local neutrophil recruitment. As the imaging experiments indicated AMs most avidly acquired the Spike protein, some interstitial macrophages also retained it, while only a low percentage of the lung neutrophils and dendritic cells were positive following isolation (*Figure 2B*). Instillation of the glycan-deficient Spike protein reduced the monocyte cellular infiltrate and decreased the % of AMs that retained the protein. The high mannose Spike protein slightly increased the numbers of neutrophil and macrophages in the lung despite a lower uptake by alveolar and interstitial macrophages. Finally,

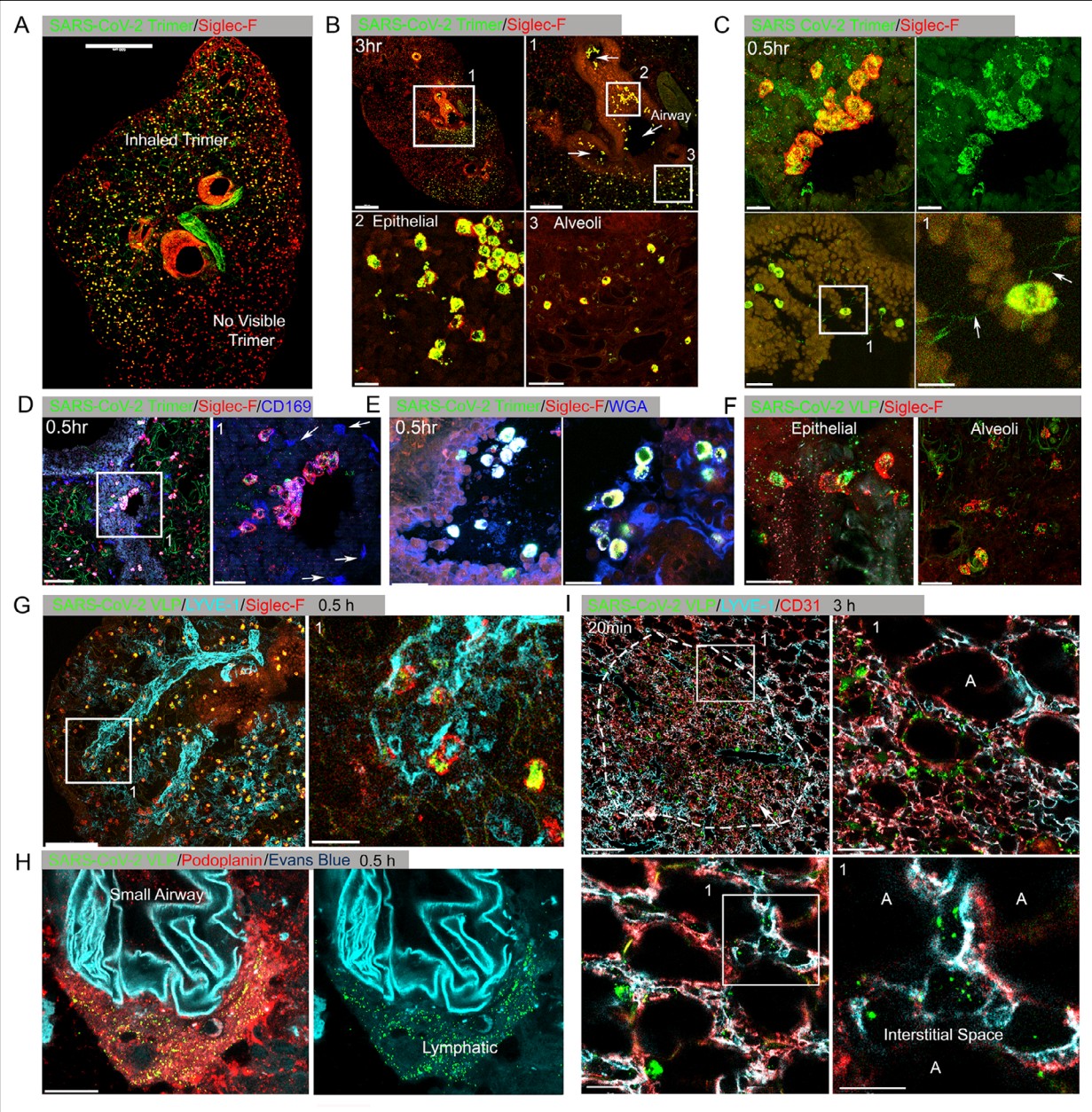

**Figure 1.** Lung images following intranasal administered SARS-CoV-2 Spike protein. (**A**) Confocal micrograph of a lung section 18 hr post SARS-CoV-2 Spike protein (Trimer) installation. Alveolar macrophages (AMs) visualized with Siglec-F antibody. Alexa Fluor 488-conjugated recombinant SARS-CoV-2 Spike protein (3 µg in 50 µl of saline) was inoculated by nasal installation. Spike-reached lung region (Instilled Trimer) and non-reached area (No visible Trimer) noted on the micrograph. Scale bar, 500 µm. (**B**) Confocal micrographs of lung collected at 3 hr post Spike protein installation. AMs in the large airway and alveoli (upper left) area is shown. ROI-1 (Box 1) (upper right) is enlarged, and arrows indicate large airways. ROI-2 (Box 2) (lower left) shows AMs bearing SARS-CoV-2 Spike protein on airway epithelial cells. ROI-3 (Box 3) (lower left) shows Spike protein bearing AMs in alveoli. Scale bars, 500, 200, 20, and 50 µm. (**C**) Confocal micrographs of lung collected at 0.5 hr post SARS-CoV-2 Spike protein installation. AMs were visualized with Siglec-F antibody (upper left and right). AMs connected to each other (lower left) via tunneling nanotubes (arrows) (lower right). Scale bars, 20, 20, 30, and 10 µm. (**D and E**) Confocal micrographs of lung collected at 0.5 hr post Spike installation. SARS-CoV-2 Spike protein (green) on airway epithelium and CD169[+] macrophages (**D**). Arrows indicate CD169[+] macrophages (right). Scale bars, 100 and 30 µm. AMs with SARS-CoV-2 Spike protein and Mucin (Alexa Fluor 647-conjugated wheat germ agglutin [WGA]) on airway epithelium (**E**). Scale bars, 30 and 20 µm. (**F**) Confocal micrographs of lung collected at 2 hr post SARS-CoV-2 Spike protein incorporated viral like particle (VLP) installation (green). AMs (red, SiglecF) on epithelium (left) and in alveoli (right). Scale bars, 20 µm. (**G**) Confocal micrographs of lung collected at 0.5 hr post SARS-CoV-2 Spike VLP (green) installation. AMs (red, Siglec-F antibody) and lung vasculatures visualized LYVE-1 (cyan) (top, left). LYVE-1[+] vasculature-associated AMs bearing VLPs is highlighted (box) and enlarged (top, right). Scale bars, 100 and 25 µm. (**H**) A confocal micrograph shows a lung lymphatic vasculature visualized with Podoplanin antibody. Fifty microliters of a mixture of Evans blue (cyan) (5 µg) and Spike bearing VLPs (green) (0.5 million counts) were applied to the mouse nose. VLPs in lymphatics associated with a small

*Figure 1 continued on next page*

*Figure 1 continued*

airway highlighted (right). Scale bars, 20 µm. (**I**) Confocal micrographs of lung collected 20 min post Spike incorporated VLP (green) instillation. Lung vasculatures visualized CD31 (red) and LYVE-1 (cyan). Damaged lung tissue is indicated (dotted line) (upper left). Border of damaged tissue and intact alveoli (box) is enlarged (upper right). A lymphatic structure stained by LYVE-1 in intact alveolus is highlighted (lower left). An image of VLPs in LYVE-1⁺ lymphatic portal in alveoli (box) is enlarged (lower right). 'A' indicates alveolus. Scale bars, 100, 20, 25, and 10 µm.

The online version of this article includes the following source data and figure supplement(s) for figure 1:

**Source data 1.** Original image of the lung section in *Figure 1A*.

**Source data 2.** Original image of the lung section in *Figure 1B* upper left.

**Source data 3.** Original image of the lung section in *Figure 1B* upper right.

**Source data 4.** Original image of the lung section in *Figure 1D*.

**Source data 5.** Original image of the lung section in *Figure 1G*.

**Source data 6.** Original image of the lung section in *Figure 1I*.

**Figure supplement 1.** Preparation of SARS-CoV-2 proteins.

**Figure supplement 1—source data 1.** Original image of the coomassie blue stain in *Figure 1—figure supplement 1*.

**Figure supplement 1—source data 2.** Original image of the coomassie blue stain in *Figure 1—figure supplement 1* (labelled).

**Figure supplement 2.** Confocal micrographs were taken for staining control and to analyze neutrophil fragmentation caused by SARS-CoV-1 Spike viral like particles (VLPs), comparing them to delta Env VLPs.

**Figure supplement 2—source data 1.** Source data for *Figure 1—figure supplement 2C*.

**Figure supplement 2—source data 2.** Original image of analized Ly6G+ granaules for delta envelop VLP in *Figure 1—figure supplement 2B*.

**Figure supplement 2—source data 3.** Original image of analized Ly6G+ granaules for Spike VLP in *Figure 1—figure supplement 2B*.

the D614G mutation reduced the cellular infiltrate compared to the unmutated protein and had a slightly different cell binding profile as a greater percentage of monocytes, eosinophils, and dendritic cells retained it (*Figure 2A and B*). We also tested whether the addition of human ACE2 (hACE2) affected the cellular uptake of the Spike protein following intranasal administration by comparing wild-type and the hACE2 transgenic mice. While we saw little difference by imaging, we did note some minor changes in the cellular uptake pattern, most notably an increase uptake by AM and a decrease uptake by interstitial macrophages (*Figure 2—figure supplement 1*).

## Altered lung vascular permeability, neutrophil recruitment, and lung damage 3 hr post instillation of the Spike protein

The increase in lung leukocytes following the Spike protein nasal instillation suggested that it may have altered the vascular permeability of the lung. To assess whether vasculature permeability changes had occurred, we intravenously injected Evans blue dye, which in the absence of a permeability defect remains confined to the vasculature (*Matthew et al., 2002*). Prior to injecting the Evans blue, we instilled in the nasal cavity the SARS-CoV-2 Spike protein, the S1 subunit of the HCoV-HKU1 Spike protein, or a saline control. Inspection of the lungs from the SARS-CoV-2 spike protein-treated mice revealed a strong increase in Evan blue staining while the lungs from the S1 subunit HCoV-HKU1-treated mice had a minimal increase. Quantifying the Evans blue dye in collected lungs and liver confirmed the increase in vasculature permeability in the lungs of the SARS-CoV-2 Spike protein-treated mice (*Figure 3A*). Next, we compared the S1 subunit of SARS-CoV-2 Spike protein to the HCoV-HKU1 S1 subunit. The S1 subunit preparations were purified from transfected HEK 293 cells. The results of administrating the S1 subunit of HCoV-HKU1 Spike protein did not differ from the saline control while the S1 subunit from SARS-CoV-2 like the SARS-CoV-2 trimer increased the lung vascular permeability to Evans blue (*Figure 3—figure supplement 1A*). We also tested whether the presence of hACE2 affected the response by using K18-hACE2 transgenic mice as the recipients. The presence of the K18-hACE2 transgene slightly increased the lung vasculature permeability upon SARS-CoV-2 trimer instillation (*Figure 3—figure supplement 1B*). Consistent with the increase in lung vascular permeability 3 hr post SARS-CoV-2 Spike protein instillation, Siglec-F-positive lung macrophages bearing the Spike protein were surrounded by neutrophils and neutrophil fragments (*Figure 3B and C*).

Next, we investigated the potential impact of modifying the glycans displayed by the spike proteins on lung vasculature homeostasis and neutrophil recruitment. Imaging fixed lung tissue from

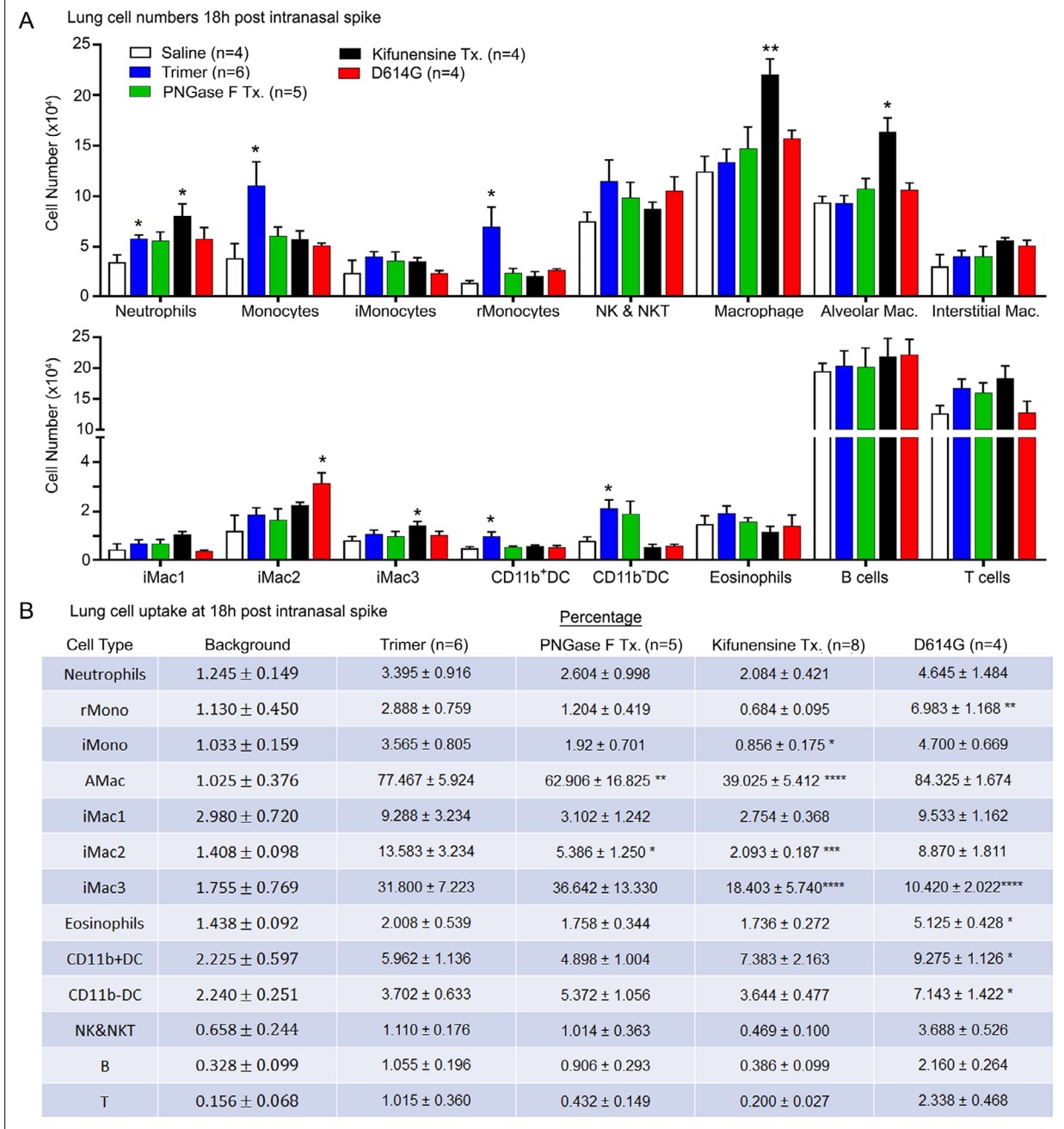

**Figure 2.** Lung leukocyte profile following intranasal administration of SARS-CoV-2 Spike proteins. (**A**) Leukocyte numbers in isolated lung tissue. Eighteen hr post intranasal administration of indicated Spike proteins lung tissue collected, processed, and leukocyte cell numbers determined by flow cytometry. Numbers of mice analyzed are indicated. Data mean ± SEM, *p<0.05; **p<0.005. (**B**) SARS-CoV-2 Spike protein uptake by lung leukocytes 18 hr following intranasal administration. Flow cytometry results from indicated mice. Values significantly different from the wild-type (WT) SARS-CoV-2 protein (Trimer) are indicated, *p<0.05; **p<0.01; ***p<0.005; ****p<0.001.

The online version of this article includes the following source data and figure supplement(s) for figure 2:

**Source data 1.** Source data for *Figure 2A*.

**Source data 2.** Source data for *Figure 2B*.

**Figure supplement 1.** Intranasal SARS-CoV-2 administration to K18-hACE2 mice.

**Figure supplement 1—source data 1.** Source data for *Figure 2—figure supplement 1B*.

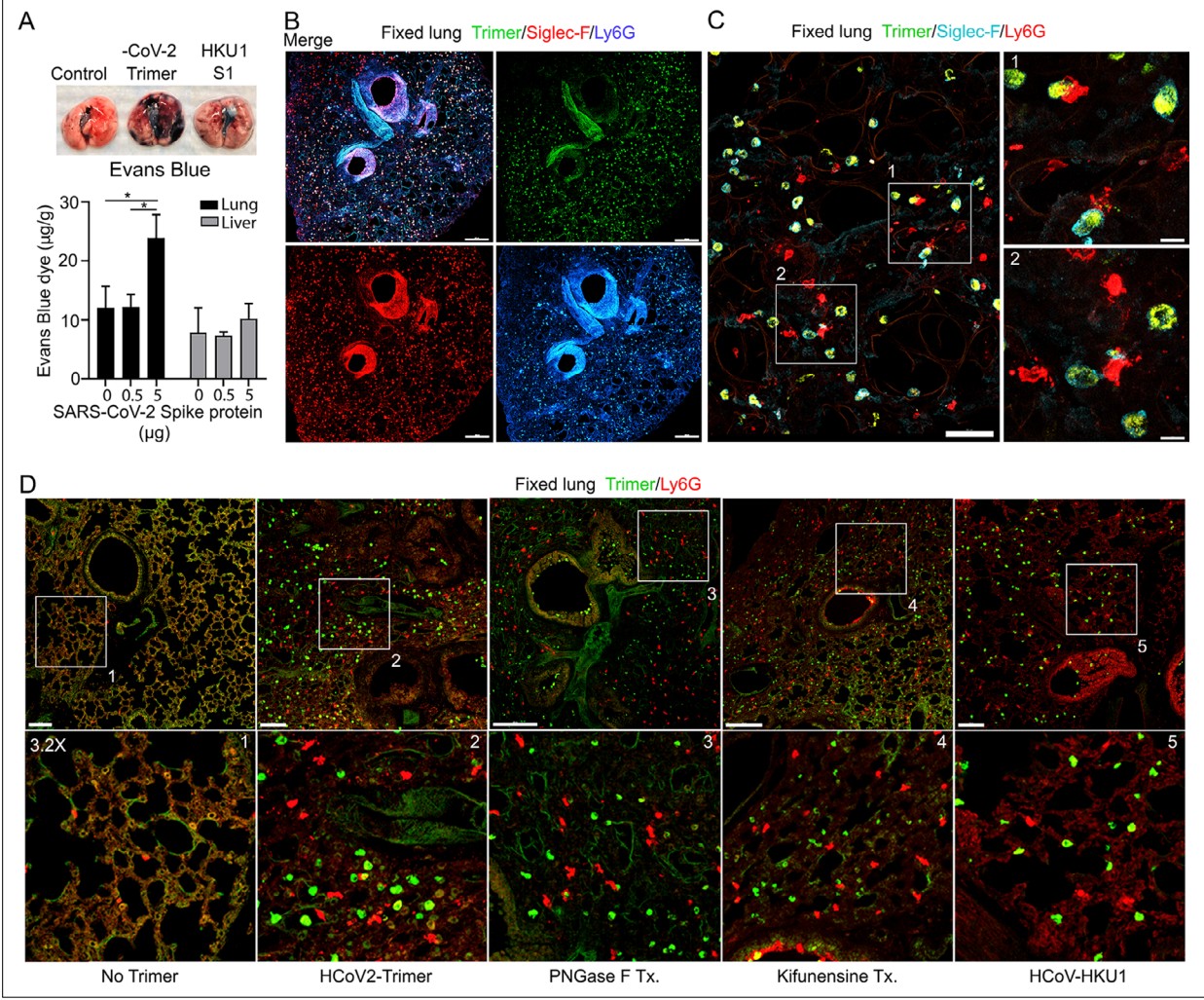

**Figure 3.** Increased lung vascular permeability and neutrophil localization following intranasal SARS-CoV-2 Spike protein. (**A**) Tissue photographs of lungs collected 2.5 hr post Spike protein (-CoV-2 Trimer), the S1 subunit of the human coronavirus (HKU1 S1), or saline instillation. Evans blue (200 µl of 5 mg/ml, PBS) was injected i.v. 1.5 hr post intranasal Spike administration. Lungs collected 1 hr after Evans blue injection. Evans blue amounts in lung and liver tissue are shown. (**B**) Confocal micrographs of lung collected at 18 hr post Spike protein. Alveolar macrophages (AMs) and neutrophils detected with Siglec-F and Ly6G antibodies, respectively. Scale bar, 500 µm. (**C**) Confocal micrographs of lung collected at 3 hr post SARS-CoV-2 Spike protein installation show SARS-CoV-2 Spike protein (green), neutrophils (red), and AMs (cyan). ROI-1 and -2 (boxes) show contact between AMs and neutrophils and are enlarged in the right panels. Scale bars, 50 and 10 µm. (**D**) Confocal micrographs of lung collected at 3 hr post each SARS-CoV-2 Spike proteins installation show SARS-CoV-2 Spike protein (green) and neutrophils (red). Left to right panels show saline (No Trimer), SARS-CoV-2 Spike protein (HCoV-2-Trimer), PNGase F-treated SARS-CoV-2 Spike protein (PNGase F Tx.), SARS-CoV-2 Spike protein purified from Kifunensine-treated cells (Kifunensine Tx.), and the S1 subunit of the human coronavirus HKU1 (HCoV-HKU1) Spike protein. ROIs in each upper panel (boxes) are enlarged (×3.2 magnification) in lower panels. Scale bars, 200 µm. Two-way ANOVA multiple comparisons, *p<0.05, (n=3). ROI, regions of interest.

The online version of this article includes the following source data and figure supplement(s) for figure 3:

**Source data 1.** Source data for *Figure 3A*.

**Source data 2.** Original image of the lung section in *Figure 3C*.

**Figure supplement 1.** Quantification of lung permeability and neutrophil recruitment.

**Figure supplement 1—source data 1.** Source data for *Figure 3—figure supplement 1A–C*.

mice instilled with saline, SARS-CoV-2 Spike protein, glycan-deficient protein, high mannose protein, and the HCoV-HKU1 S1 subunit recombinant protein revealed enhanced neutrophil recruitment with each SARS-CoV-2 Spike protein preparation. We analyzed the neutrophil count in a 200×200 µm² area at six different locations. The surveyed areas of the SARS-CoV-2 Spike protein instilled mice had approximately a 6.5-fold higher neutrophil density compared to the saline-treated control mice. The

high mannose version of the Spike protein recruited fewer neutrophils at the 3 hr time point while the PNGase F-treated version did not differ from the untreated trimer (*Figure 3D*, *Figure 3—figure supplement 1C*).

## Neutrophil recruitment and damage in the cremaster muscle following local protein injection, and in the liver following intravenous injection

An exteriorized cremaster muscle is commonly used to intravitally image mouse neutrophil transmigration and interstitial motility (*Yan et al., 2021*). Although this model lacks Siglec F-positive

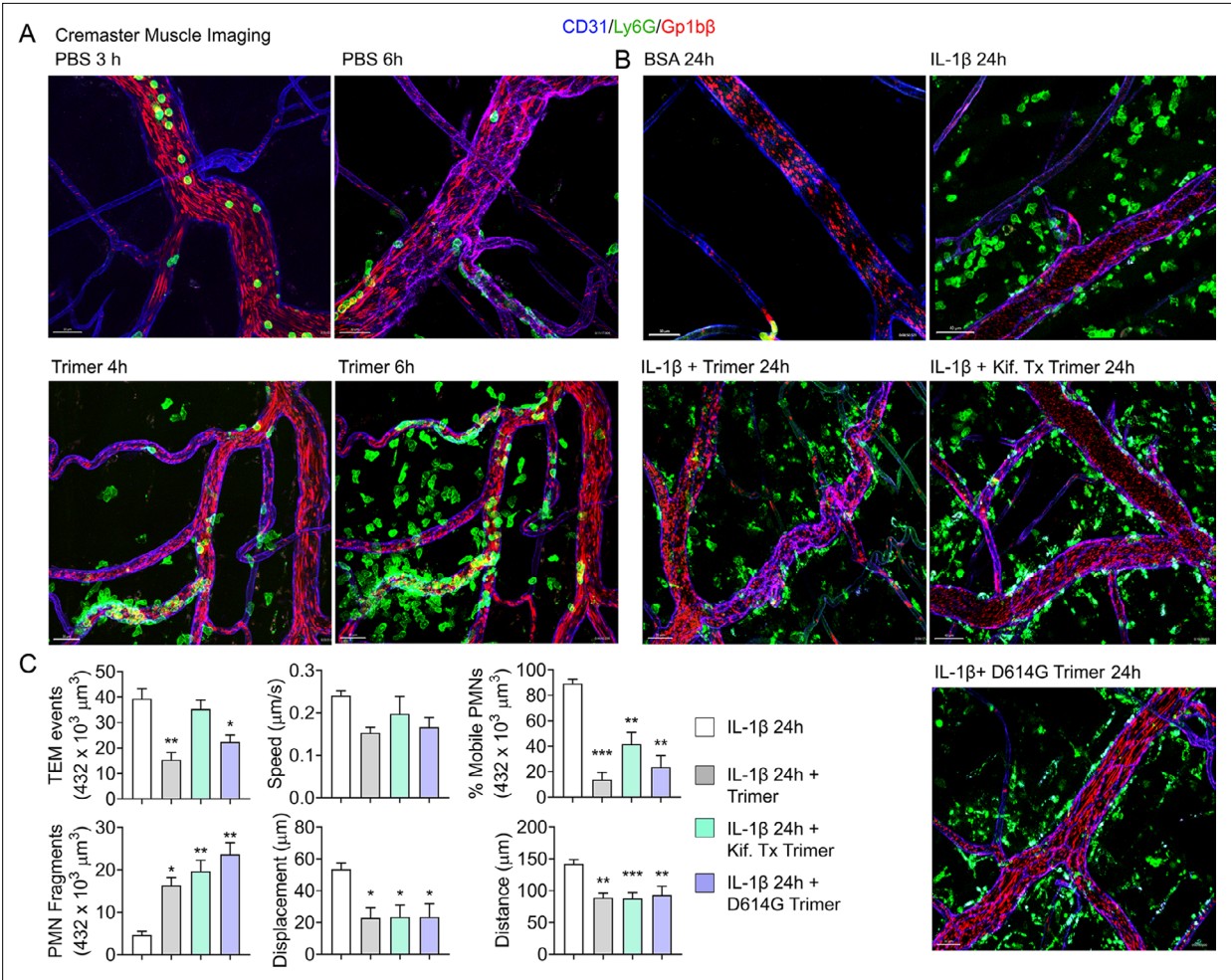

**Figure 4.** Mouse neutrophil damage following intrascrotal injections of Spike proteins. (**A and B**) Confocal snapshots of time-lapse movie of blood vessels in the cremaster muscle following PBS, bovine serum albumin (BSA), Spike protein preparations (Trimer), IL-1β, or IL-1β plus Spike protein (5 μg) at indicated time points. Fluorescently labeled CD31, Ly6G, and Gp1bβ outlined the blood vessel endothelium, neutrophils, and platelets, respectively. Scale bars, 30 μm, except far right top and middle panels, 40 μm. (**C**) Analysis of imaging data. The count of transendothelial migration (TEM) occurrences, neutrophil (PMN) fragments, and the percentage of mobile neutrophils examined within a defined imaging volume over a 20-minute period. Results of neutrophil tracking in the same defined volume including speed, displacement, and distance. Imaris software used for the tracking. *p<0.05; **p<0.005.

The online version of this article includes the following source data, source code, and figure supplement(s) for figure 4:

**Figure supplement 1.** Quantification of neutrophil count in cremaster muscle.

**Figure supplement 1—source data 1.** Source data for *Figure 4—figure supplement 1*.

**Figure supplement 2.** Localization of Spike protein in the liver following intravenous administration.

**Figure supplement 3.** Localization of SARS-CoV-2 Spike protein at other sites following intravenous injection and subcutaneous injection near the inguinal lymph node.

**Figure supplement 3—source code 1.** Original image of the spleen in *Figure 4—figure supplement 3A*.

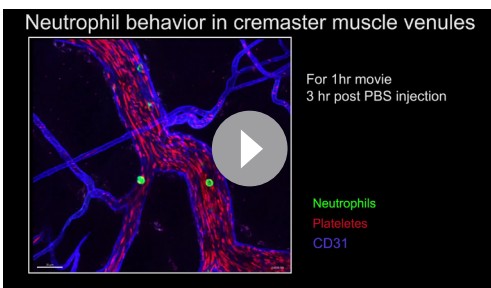

**Video 1.** Local Spike protein injection results in neutrophil recruitment in the cremaster muscle. Confocal intravital microscopy movie shows neutrophils (green), platelets (red), endothelium (blue) immunostained with Alexa Fluor 488-Gr1, DyLight 649-GP1bβ, and Alexa Fluor 555-CD31 monoclonal antibodies, respectively. Intrascrotal injection of PBS or Spike protein (5 μg) 3–4 hr prior to imaging. The first sequence spans an ~60 min (PBS) while the second sequence spans an ~45 min (Spike) imaging period. Images were captured at ~1 frame per 30 s with ~×100 magnification. Time counter is hour:minute:second.
https://elifesciences.org/articles/86764/figures#video1

macrophages, it is worth monitoring the effect of the SARS-CoV-2 Spike protein on neutrophils recruited into an inflammatory site. Within 3 hr local IL-1β injection recruits bone marrow neutrophils and leads to their transmigration into the nearby tissue. Co-injecting fluorescently labeled Ly6G and Gp1bβ antibodies identified neutrophils and platelets, respectively. Local PBS injection and intermittent imaging over a 6 hr time frame revealed occasional blood neutrophil and numerous flowing platelets (*Figure 4A*, *Video 1*). Local intrascrotal injection of unlabeled SARS-CoV-2 Spike protein caused a modest recruitment of neutrophils at 4–6 hr post injection (*Figure 4A*, *Figure 4—figure supplement 1*, *Video 1*). To better assess the long-term effect of the SARS-CoV-2 Spike protein, we co-injected IL-1β and waited until the following day to image. Typically, at 24 hr post IL-1β the inflammatory response is resolving, and the interstitial neutrophil numbers are declining from their peak (*Video 2*). In contrast, each of the Spike protein preparations adversely affected neutrophil motility and morphology. Numerous neutrophils and neutrophil fragments localized along the blood vessel walls and were scattered within the interstitium (*Figure 4B*, *Video 2*). We did not note any significant difference between the different SARS-CoV-2 Spike protein preparations as each caused neutrophil fragmentation and a decline in mobile neutrophils (*Figure 4C*).

We also assessed the impact of injecting the labeled SARS-CoV-2 protein into the blood on the liver sinusoids using intravital microscopy. Snapshots at 3 hr post infusion revealed Spike protein outlined liver sinusoid endothelial cells co-localized with the CD31 delineated sinusoid endothelial membranes. Kupffer cells identified by F4-80 immunostaining rapidly acquired large amounts of the infused Spike protein (*Figure 4—figure supplement 2A*, *Video 3*). The sinusoids, normally devoid of neutrophils, contained many Ly6G⁺ neutrophils. However, by 18 hr post infusion the amount of Spike protein outlining the sinusoids had declined as had the Kupffer cell-associated material (*Figure 4—figure supplement 2B*). The neutrophil infiltration had largely resolved suggesting that the inflammatory signals had declined. The origin of the white dots interspersed between the sinusoids is unknown, but they were not present at 3 hr post infusion despite identical imaging conditions. One possibility is that altered vascular permeability had allowed some labeled antibodies to leak into the liver parenchyma. To determine whether other endothelial beds also acquired intravenous administered Spike protein, we examined the spleen, heart ventricle, and Peyer's patches. At 3 hr post infusion endothelial cells at these three sites had acquired the labeled Spike proteins, most prominently in Peyer's patches (*Figure 4—figure supplement 3A–C*). Finally, we intravitally imaged the inguinal lymph node at 3 and 18 hr after local injection. Typically, locally injected material rapidly enters nearby afferent lymphatics for delivery to the lymph node where subcapsular sinus macrophages first encounter it (*Park et al., 2015*). If it bypasses these macrophages

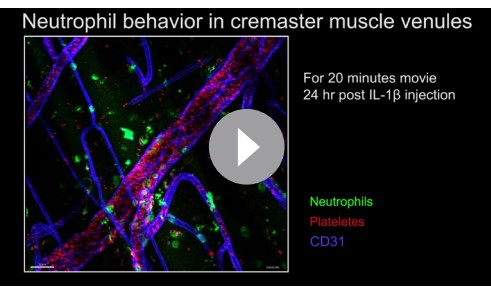

**Video 2.** Local injection of Spike proteins causes neutrophil fragmentation. Confocal intravital microscopy movie shows neutrophils (green), platelets (red), endothelium (blue) immunostained with Alexa Fluor 488Gr1, Dylight 649-GP1bβ, and Alexa Fluor 555-CD31 monoclonal antibodies, respectively. Intrascrotal injection of Il-1β or Il-1β plus 5 μg Spike protein ~20 hr prior to imaging. Both sequences span a 20 min period. Images were captured at ~1 frame per 30 s with ~×100 magnification. Time counter is hour:minute:second.
https://elifesciences.org/articles/86764/figures#video2

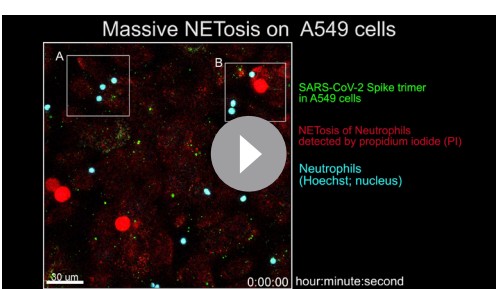

**Video 3.** Intravenously injected Spike protein outlines liver sinusoids and accumulates on Kupffer cells. The liver imaging was taken at three different time points; from 20 min to 1 hr, from 1 hr 10 min to 1 hr 15 min, and from 1 hr 25 min to 46 min after SARS-CoV-2 Spike protein (green, Alexa Fluor 488) injection. An image sequence of a 12 µm z-projection was acquired with 40× lens as scanning speed of 11.05 s between frames. Kupffer cells (magenta, F4/80) and neutrophils (red, Ly6G) in liver sinusoid vasculature (cyan, CD31) were visualized with antibody injection into tail vein at 30 min prior to imaging. The scale bar represents 20 µm. Time counter is hour:minute:second.

https://elifesciences.org/articles/86764/figures#video3

lymph-borne material flows into the lymph node medullary region, where medullary macrophages reside (*Figure 4—figure supplement 3D*). At both the 3 hr time point (data not shown) and at 18 hr subcapsular macrophages showed little interest in the Spike protein, but medullary macrophages avidly acquired it. In contrast to the liver, we did not detect neutrophil recruitment into the lymph node at either time point after tail base injection.

## Murine and human neutrophils NETosis following exposure to the SARS-CoV-2 Spike protein

Live cell imaging of lung sections following intra-nasal instillation of the Spike protein revealed ongoing neutrophil damage and likely NETosis (*Figure 5A*). Time-lapse images show several disrupted neutrophils near a Siglec-F-positive AM along with Spike protein bound to a dying neutrophil. This observation along with the cremaster muscle imaging suggested direct neutrophil toxicity. Furthermore, a previous study had shown that Spike protein induced neutrophil NETosis (*Youn et al., 2021*). To confirm that the Spike protein can trigger neutrophil NETosis and to compare different Spike protein preparations, we briefly exposed murine bone marrow neutrophils and assessed the % of cells undergoing cell death using a flow-based assay. Exposure to *N*-formylmethionyl-leucyl-phenylalanine (fMLP) served as a positive control. The lowest concentration of Spike protein (0.1 µg) tested increased the number of dying neutrophils (*Figure 5B*). Switching to human neutrophils purified from human peripheral blood, exposure to tumor necrosis factor-α (TNF-α), LPS, or phorbol myristate acetate (PMA) for 4 hr or overnight reduced their viability as expected. Addition of SARS-CoV-2 or the D614G mutant protein (1 µg/ml) decreased their viability at 4 hr and more dramatically upon overnight exposure. The high mannose versions of the Spike protein and D614G Spike had a slightly greater toxicity at 4 hr but had a similar impact in the overnight assay (*Figure 5C*). Finally, we imaged human neutrophils overlaid on SARS-CoV-2 Spike protein-treated A549 cells, a human lung epithelial cell line. Numerous dying neutrophils could be observed likely undergoing NETosis (*Figure 5D*, *Video 4*). These results confirm that the SARS-CoV-2 Spike protein can cause neutrophil damage potentially exacerbating the inflammatory response.

## Human peripheral blood monocytes, B cells, neutrophils, and dendritic cells bind the SARS-CoV-2 Spike protein

Next, we assessed human peripheral blood leukocyte binding using the fluorescently labeled Spike protein. Binding assays were performed on ice to

**Video 4.** Neutrophils undergo NETosis when plated on A549 cells in the presence of Spike protein. A549 cells were plated on chamber slide 48 hr before trimer treatment. SARS-CoV-2 Spike trimer (green) was added to A549 cell culture 24 hr before neutrophil seeding. Purified human neutrophils stained with Hoechst (cyan) and were seeded on A549 cells in propidium iodide (PI, 1 µg/ml) containing culture media. NETosis of neutrophils were detected by exposed DNA (red, PI). An image sequence of a 20 µm z-projection was acquired with 40× lens at a scanning speed 1 frame/10 min over 5 hr. Regions of interest (Box A and Box B) demonstrate that a typical NETosis of neutrophil contacting on trimer bearing A549 cell was enlarged in second part of video. Scale bars, 30 and 10 µm. Time banner, hour:minute:second.

https://elifesciences.org/articles/86764/figures#video4

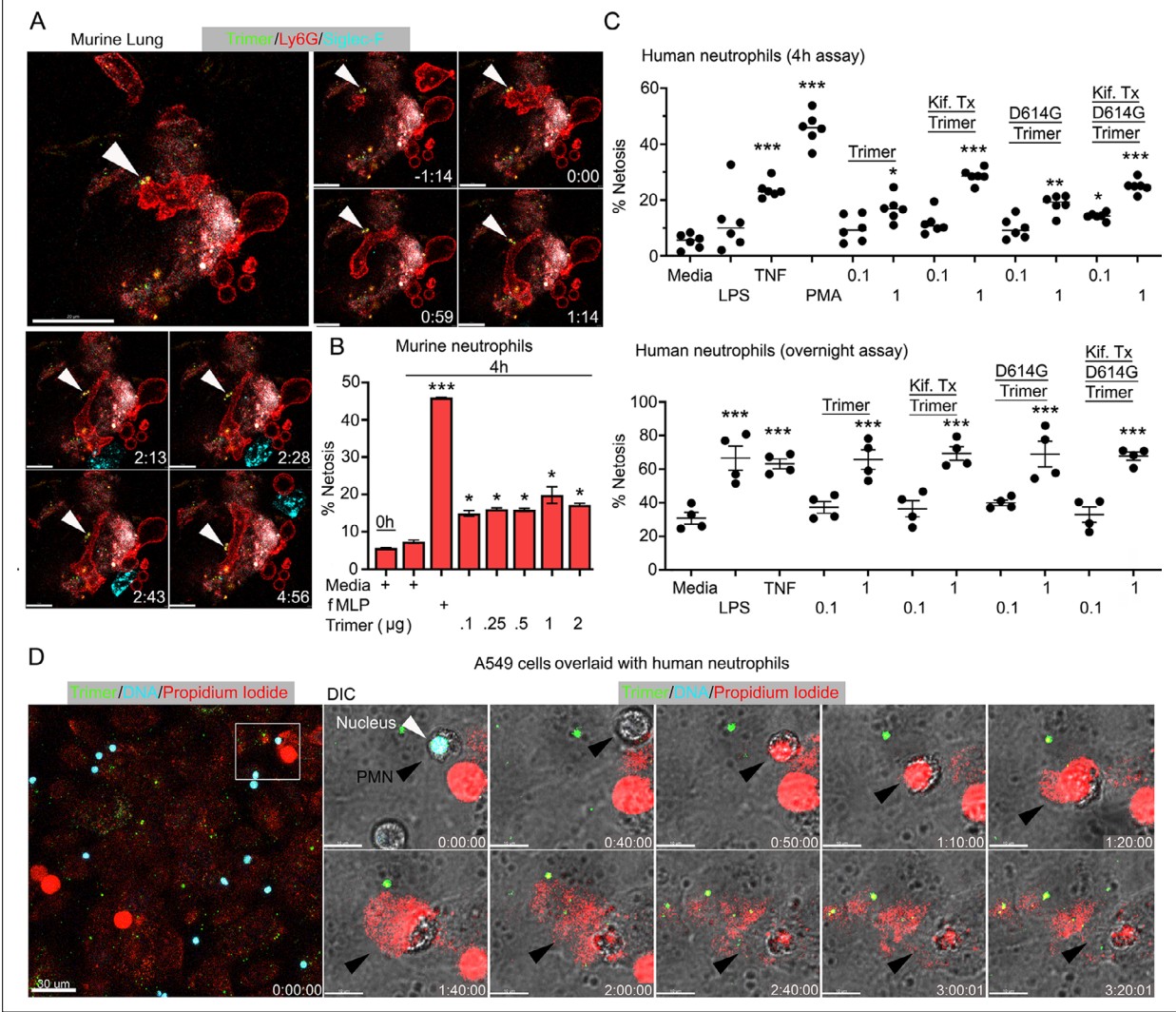

**Figure 5.** Neutrophil injury following exposure to SARS-CoV-2 Spike proteins. (**A**) Time-lapse images of a freshly sliced lung section 3 hr after SARS-CoV-2 Spike protein instillation. Neutrophils (Ly6G) and alveolar macrophages (AMs) (Siglec-F) were visualized by injected fluorescently tagged antibodies. Serial images show neutrophil behavior on Spike protein (Trimer) bearing cells. Arrowheads indicate Spike protein deposition. The time banner is set to 0:00 as neutrophil contacts Spike protein. Scale bars, 100, 25, and 20 μm. (**B**) Purified murine bone marrow neutrophils were exposed to increasing concentrations of Spike protein or fMLP (*N*-formylmethionyl-leucyl-phenylalanine) for 4 hr or not. The percentage of neutrophils undergoing NETosis was measured by flow cytometry (n=2). (**C**) Purified human peripheral blood neutrophils were exposed to various Spike protein preparations, lipopolysaccharide (LPS), tumor necrosis factor-α (TNF), or phorbol myristate acetate (PMA). Graphs show the percentage of DAPI⁺Helix NP NIR⁺ cells after either 4 hr (n=6) or overnight culture (n=4). Each point represents a different donor. (**D**) A still image of in vitro time-lapse movie shows exposed neutrophil DNA (red, propidium iodide), neutrophil nucleus (cyan, Hoechst), and SARS-CoV-2 Spike protein (trimer, green) bearing A549 cells. Plated A549 cells were treated with Spike protein (0.5 μg/ml) 24 hr prior to Hoechst-stained human neutrophil seeding. Propidium iodide (1 μg/ml) was added to the culture media and time-lapse images acquired every 10 min for 5 hr. Sequential DIC images overlaid with fluorescent images show neutrophil NETosis. White arrowhead indicates a neutrophil's intact nucleus (cyan, Hoechst). Black arrowheads delineate a neutrophil undergoing NETosis as detected by exposed DNA (red). Scale bars, 30 and 10 μm. *p<0.05; **p<0.01; ***p<0.005.

The online version of this article includes the following source data for figure 5:

**Source data 1.** Source data for *Figure 5B*.

**Source data 2.** Source data for *Figure 5C*.

avoid endocytosis using Hanks' Balanced Salt Solution (HBSS) with added Ca²⁺ and Mn²⁺, or with EDTA to assess cation dependence. Representative CD4 T cell, B cell, monocyte, or neutrophil flow patterns using either the Spike protein or the high mannose version are shown (*Figure 6A*). The flow cytometry results demonstrated that cations enhanced leukocyte binding particularly so with

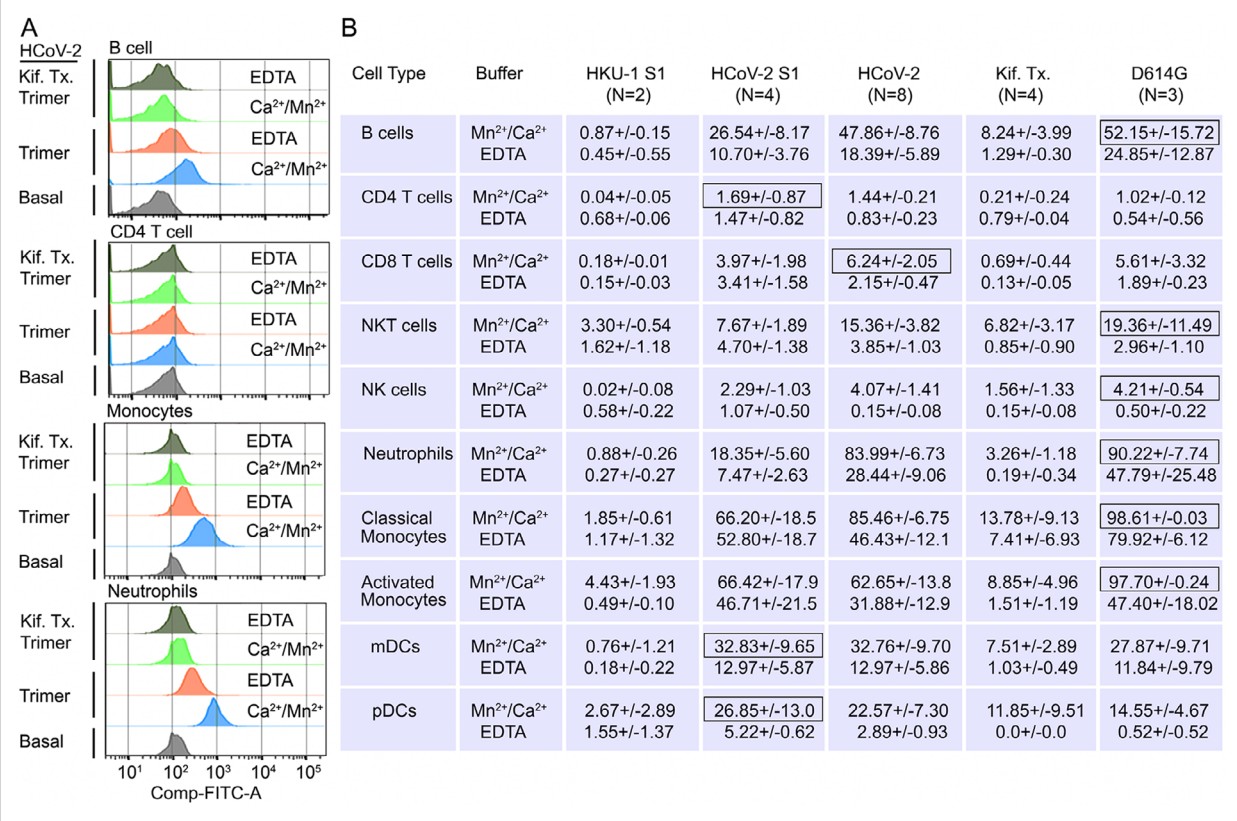

**Figure 6.** Binding of recombinant SARS-CoV Spike proteins to human peripheral blood leukocytes. (**A**) Representative flow cytometry histograms of various cell populations prepared from whole blood incubated with labeled recombinant SARS-CoV-2 Spike protein (Trimer), or Spike protein from Kifunensine-treated cell (Kif. Tx. Trimer), binding done in the presence of $Ca^{2+}/Mn^{2+}$ or EDTA. (**B**) Percentage of various cell populations prepared from whole blood that bound labeled HKU1 S1 protein (HKU-1 S1), SARS-CoV-2 S1 protein (HCoV-2 S1), SARS-CoV-2 Spike protein (HCoV-2 Trimer), Spike protein purified from Kifunesine-treated cells (Kif. Tx. Trimer), or D614G Spike protein. Binding done in the presence of $Ca^{2+}/Mn^{2+}$ or EDTA and assessed by flow cytometry. Background fluorescent (unstained) subtracted from fluorescent signal. Data are from two to eight independent experiments. The recombinant protein that bound the highest % of cells of the different cell types is designated with black outline.

the high mannose protein. The SARS-CoV-2 S1 protein bound better than did the HKU1 S1 protein, most evident with NK cells, neutrophils, monocytes, and dendritic cells (*Figure 6B*). The SARS-CoV-2 Spike protein bound better than did the S1 protein with B cells, neutrophils, and monocytes exhibiting the best binding. The high mannose protein bound less well to the different leukocyte subsets. Surprisingly CD4 T cells poorly bound each of the proteins, irrespective of the presence or absence of cations. The D614G Spike protein bound better to neutrophils and monocytes than did the wild-type Spike protein (*Figure 6B*).

## Murine and human cells use Siglecs to help capture the SARS-CoV-2 Spike Protein

Since murine and human leukocytes lack significant ACE2 levels, the major entry receptor for SARS-CoV-1 and -2, they likely use other receptors to capture the Spike proteins. The strong co-localization with Siglec-F expressing murine AMs prompted an examination of the role of Siglec-F and other Siglecs in capturing the Spike proteins. Siglecs are transmembrane proteins that exhibit specificity for sialic acids attached to the terminal portions of cell surface glycoproteins. Several viruses take advantage of sialic acid-Siglec interactions for cell targeting, spreading, and trans-infection (*Hammonds et al., 2017*; *Perez-Zsolt et al., 2019*; *Perez-Zsolt et al., 2021*). Initially, we established a bead assay to assess the binding of SARS-CoV-2 Spike proteins. We coupled the S1 domain protein, the SARS-CoV-2-stabilized Spike protein, or the PNGase F-treated Spike protein and reacted the beads with fluorescently labeled antibody or different recombinant proteins. Flow cytometry analysis revealed strong binding of the Spike protein antibody, hACE2, but not murine ACE2 as expected (*Figure 7A*).

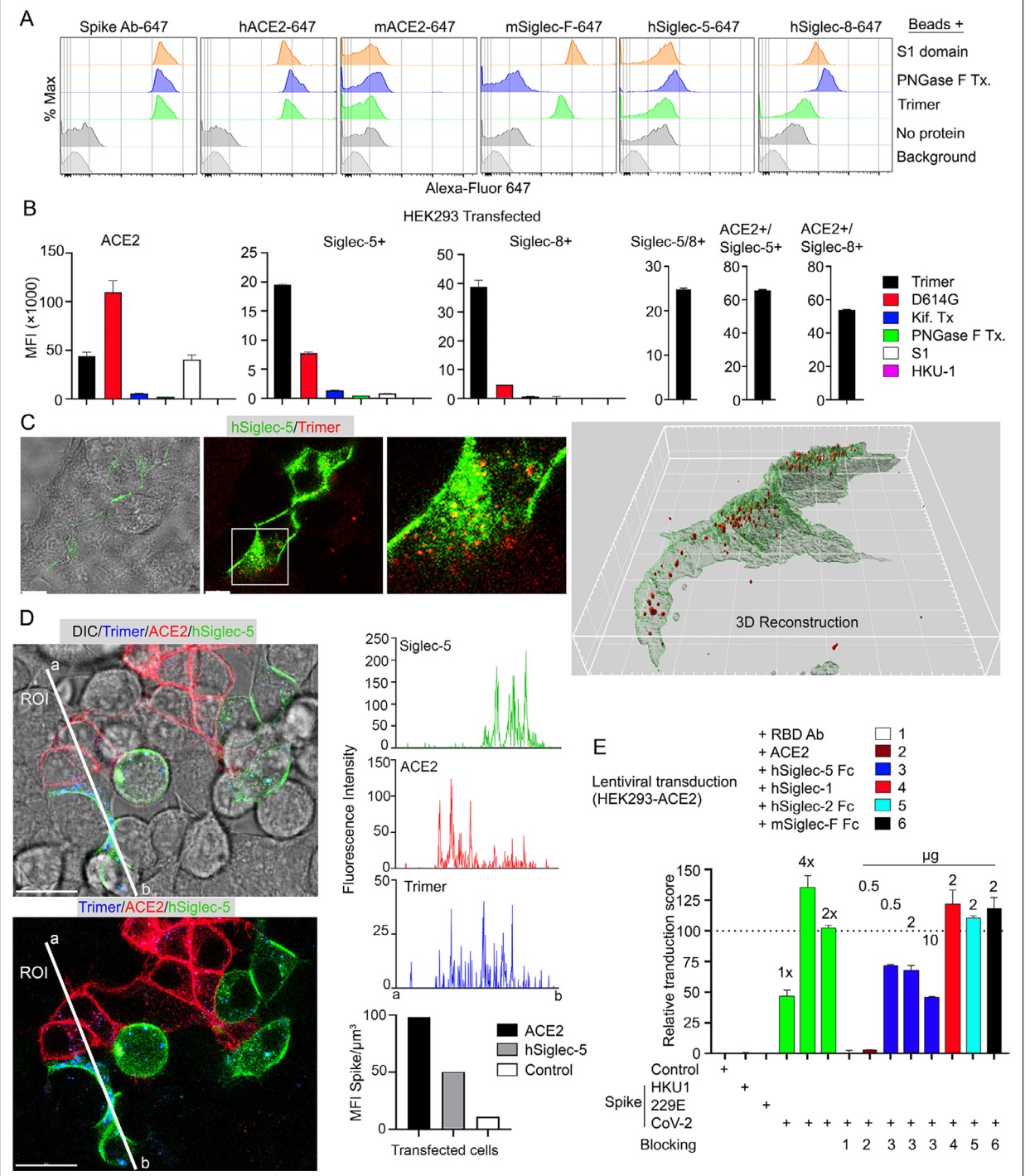

**Figure 7.** Role of Siglecs in SARS-CoV-2 Spike capture. (**A**) Histograms show mean fluorescence of conjugated recombinant proteins and antibody on nanobeads that are coated with indicated SARS-CoV-2 Spike proteins. Background signal (weak gray) measured with uncoated beads. No protein (gray) means that uncoated beads were incubated with indicated labeled recombinant protein or antibody. (**B**) Flow cytometry assessment of indicated Spike proteins binding to HEK293 cell permanently transfected with angiotensin converting enzyme II (ACE2), various Siglecs, or a combination. Dual transfected HEK293 cells noted as Siglec-5/8+, ACE2/Siglec-5+, and ACE2/Siglec-8+. Labeled recombinant proteins were incubated with $5 \times 10^3$ HEK293 cells on ice for 30 min. Data shown as mean fluorescent intensity. (**C**) DIC and confocal micrographs showing SARS-CoV-2 Spike (Trimer, red) protein acquisition by human Siglec-5-GFP transfected HEK293. ROI-1 in the middle panel is enlarged at the right panel. A 3D-reconstituted volume image of Siglec-5 transfected cell (far right) shows Siglec-5-mediated Spike protein acquisition. Scale bars, 30 and 10 μm. (**D**) DIC and confocal images show SARSCoV-2 Spike protein acquisition by human Siglec-5-GFP or ACE2-OFP transfected HEK293. Equal numbers of stable transfectants expressing Siglec-5-GFP or ACE2-OFP, and non-transfectant were seeded together. Spike protein (Trimer, blue) was added 1 hr prior to imaging. In a region of

*Figure 7 continued on next page*

*Figure 7 continued*

interest (ROI) line ab fluorescence intensity of Siglec-5-GFP, ACE2-OFP, and SARS-CoV-2 Spike trimer were analyzed. Fluorescent intensity graphs show fluorescence intensity of each signal in ROI. Amount of 3D-reconstructed volume of SARS-CoV-2 Spike protein in each stable cell and HEK293 cell were analyzed and plotted in graph. Graphs show quantity of mean fluorescence intensity of SARS-CoV-2 Spike protein in unit volume ($\mu m^3$) of indicated cell types. Scale bars, 20 µm. (**E**) Transduction of Spike protein expressing lentiviruses into ACE2/HEK293 transfectants. Lentiviruses envelop proteins: control (none), human coronavirus HKU1 Spike protein (HKU1), human coronavirus 229E Spike protein (229E), and SARS-CoV-2 Spike protein (CoV-2) are indicated. The volume of virus concentrates used for transduction is denoted as 1×, 2×, and 4× on the graph (bright green bars). Transduction scores were normalized to 2× transduction efficiency. Various recombinant proteins or antibodies used for inhibition assay are indicated. Recombinant proteins or RBD neutralizing antibody were added 0.5 hr before lentivirus transduction. Three days later GFP expression was quantitated by flow cytometry. Results are from three separate experiments. Statistics, 2× vs 2×+hSiglec-5 Fc (10 µg), p<0.0005 (n=3). GFP, green fluorescent protein.

The online version of this article includes the following source data for figure 7:

**Source data 1.** Source data for *Figure 7B*.

**Source data 2.** Source data for *Figure 7D*.

**Source data 3.** Source data for *Figure 7E*.

**Source data 4.** Original image of a human Siglec-5-GFP transfected HEK293 cell in *Figure 7C*, 3D reconstruction.

Siglec-F also bound well, while the human Siglec-5 and Siglec-8 bound poorly despite being the structural and functional equivalents of Siglec-F, respectively (*Connolly et al., 2002*). Of note the PNGase F-treated fraction we used likely retained some N-linked glycans as it remained able to bind ACE2 although it lost Siglec-F binding (*Figure 7A*).

Despite the poor binding to the recombinant human Siglecs in the bead assay, we elected to test them in context of a human cell by expressing them in HEK293 cells. We chose HEK293 cells as they exhibited little Spike protein binding. We established ACE2, Siglec5, Siglec-8, Siglec-5/8, ACE2/Siglec-5, and ACE2/Siglec-8 expressing cell lines (*Figure 7B*). The ACE2 transfected cells behaved as anticipated binding the original Spike protein and even better the D614G version; and binding the S1 protein. The high mannose and glycan-deficient Spike proteins bound poorly. In contrast to the bead assay, both the Siglec-5 and the Siglec-8 expressing HEK293 cells bound the Spike protein, although less efficiently than did the ACE2 expressing cells. They bound the D614G version less well, and very weakly bound the S1 domain protein. The high mannose and glycan-deficient Spike proteins exhibited little binding to the Siglec expressing cells. Co-expression of Siglec-5 and -8 did not improve binding, although the ACE2/Siglec co-expressing cells bound the SARS-CoV-2 Spike protein slightly better than did ACE2-only cells. We also imaged the Siglec-5 expressing HEK293 with labeled protein (*Figure 7C*). The imaging revealed efficient uptake and Spike protein endocytosis. Co-cultured ACE2 and Siglec-5 expressing cells both captured the Spike protein while HEK293 cells not expressing either protein failed to bind or uptake it (*Figure 7D*, *Video 5*). To determine whether Siglec-5 or Siglec-8 could contribute to viral transduction, we produced lentiviral particles expressing the Spike protein from SARS-CoV-2, HKU1, or 229E and attempted to transduce the HEK293 ACE2 expressing cells. Successful transduction resulted in GFP expression, which we quantitated by flow cytometry and only occurred with the lentiviral particles with SARS-CoV-2 Spike protein incorporation (*Figure 7E*). Both a receptor binding domain antibody and recombinant ACE2 blocked transduction. Recombinant human Siglec-5 partially inhibited although it required relatively high concentrations, while human Siglec-1, human Siglec-2, and murine Siglec-F had no impact on the transduction frequency.

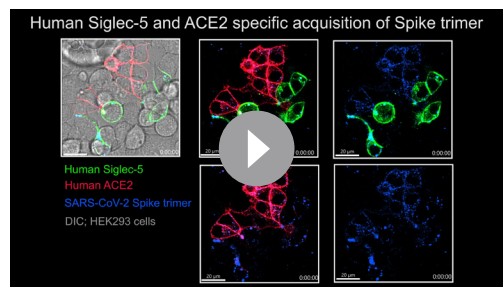

**Video 5.** hSiglec-5 expressing HEK293 cells bind and endocytosis Spike protein. Individually established stable transfected cells expressing Siglec-5-GFP or ACE2-OFP were plated in the same chamber slide with non-transfection cells. After overnight culture SARS-CoV-2 Spike protein (1 µg/ml) was overlaid for 1 hr before imaging. An image sequence of a 3 µm z-projection was acquired with 40× lens at a scanning speed of 23 s between frames. Signals visualized as SARS-CoV-2 Spike trimer (blue), Siglec-5-GFP (green), ACE2-OFP (red), and non-transfected HEK293 cells (gray). The scale bar represents 20 µm. Time counter is hour:minute:second. ACE2, angiotensin converting enzyme II. GFP, green fluorescent protein.

https://elifesciences.org/articles/86764/figures#video5

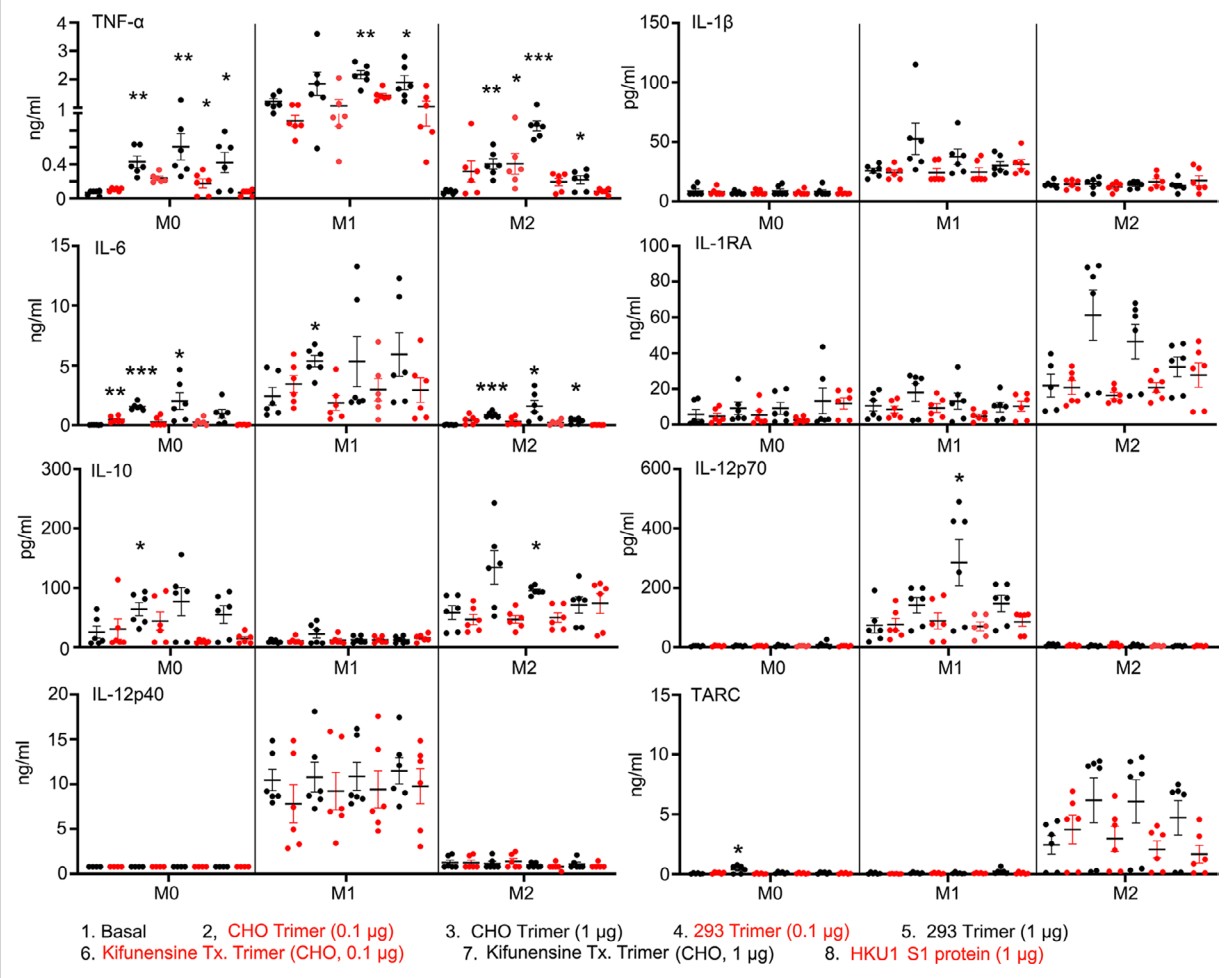

**Figure 8.** Macrophage cytokine profiles following exposure to different Spike proteins. Quantification of macrophage-associated cytokines in response to indicated Spike proteins. The graph shows the tumor necrosis factor-α (TNF-α), IL1β, IL-6, IL-12p70, IL-12p40, and TARC quantification of M0, M1, or M2 macrophage cultures following stimulation, or not, with SARS-CoV-2 Spike proteins derived from CHO cells (CHO Trimer; 0.1 or 1 μg/ml), or 293F cells (293 Trimer; 0.1 or 1 μg/ml), or purified Spike protein from Kifunensine-treated CHO cells (Kifunensine Tx. Trimer; 0.1 or 1 μg/ml), or human coronavirus HKU1 S1 protein (HKU1 S1 protein; 1 μg/ml) for 48 hr. The error bars denote the mean ± SEM. Two-way ANOVA was used to compare treated samples to basal M0, M1, or M2 population (n=6, *p<0.05; **p<0.01; ***p<0.001).

The online version of this article includes the following source data for figure 8:

**Source data 1.** Source data for *Figure 8*.

## Increased TNF-α and IL-6 production by human macrophages exposed to the SARS-CoV-2 Spike proteins

Dysregulated cytokine production contributes to the pathogenesis of severe COVID-19 infections (*Vora et al., 2021*; *Sefik et al., 2022*). To assess whether the Spike proteins can affect macrophage cytokine profiles, we cultured human monocytes using conditions that generate M0, M1, or M2 macrophages and treated them with different Spike protein preparations. We collected cell supernatants 2 days later and measured a panel of known macrophage-derived cytokines by ELISA (*Figure 8*). We found little induction of IL-1β in the cell supernatant indicating that the Spike proteins alone did not trigger the inflammasome activation in these macrophages. In contrast, we did observe significant increases in TNF-α and IL-6 secretion by the original SARS-CoV-2 Spike protein-treated macrophages irrespective of whether it was produced in CHO or HEK293F cells. The spike protein produced in HEK293F is predicted to contain more complex carbohydrates than that expressed in CHO cells. While both preparations increased IL-6 and TNF-α production, the spike protein produced by HEK293F cell elicited more TNF-α and less IL-6 compared with the CHO-produced spike protein. The spike protein

produced in CHO cells treated with Kifunesine elicited a similar cytokine profile as did the spike protein produced in CHO cells in the absence of Kifunesine. The addition of the HKU1 S1 protein did not modify cytokine production by any of the macrophage subsets.

## Discussion

This study analyzed the cell tropism of the SARS-CoV-2 Spike protein and some of its variants using fluorescently labeled proteins, flow cytometry, and mice to assess in vivo binding. The intranasal administration of SARS-CoV-2 Spike protein to mice led to its rapid uptake by Siglec-F-positive AMs and an increase in the number of neutrophils, monocytes, and dendritic cells in the lung 18 hr after instillation. Modifying the carbohydrate content of the Spike protein or using a Spike protein with a D614G mutation slightly altered the cell recruitment pattern. The high mannose Spike protein purified from the Kifunensine-treated CHO cells increased the number of lung macrophages at 18 hr despite a lower % of macrophages having retained it. As AMs express the mannose receptor (MR, CD206), a member of the C-type lectin (CLEC) family, perhaps the high mannose Spike protein is more rapidly endocytosed and degraded accounting for a lower percentage of cells retaining it. This receptor binds high mannose structures present on the surface of pathogens helping to neutralize them by phagocytic engulfment. Other cell types including immature dendritic cells, and endothelial cells in hepatic, splenic, lymphatic, and dermal microvasculature express the MR and would be expected to be targeted by the high mannose Spike protein. We confirmed Siglec-F as a capturing receptor, as it efficiently bound to the S1 domain and the SARS-CoV-2 Spike protein. Yet AMs employ additional receptors to capture the Spike protein as they still accumulated the PNGase F-treated Spike protein, which had lost binding to Siglec-F.

Arguing against a role for mSiglec-1, CD169$^+$ lung macrophages and CD169$^+$ subcapsular sinus macrophages failed to accumulate the Spike protein in vivo. The closest human paralog of mouse Siglec-F is hSiglec-8 (*Aizawa et al., 2003*). While expressed on human eosinophils and mast cells, human AMs apparently lack it. In contrast, human AMs do express Siglec-5 (*Connolly et al., 2002*). Along with its paired receptor, hSiglec-14, Siglec-5 can modulate innate immune responses (*Tsai et al., 2020*). When tested in a bead binding assay, in contrast to Siglec-F, neither hSiglec-5 or -8 bound the recombinant Spike protein, yet their expression in a cellular context allowed binding. Additionally, recombinant hSiglec-5 partially inhibited the transduction of Spike protein bearing VLPs into ACE2 expressing HEK293 cells. Evidently, the recombinant protein binding assay we employed did not fully capture the properties of hSiglec-5 or hSiglec-8 in a cellular context. A recent study of human lungs revealed scant alveolar ACE2 expression, highlighting the importance of alternative Spike protein receptors (*Hönzke et al., 2022*). Ex vivo infected human lungs and COVID-19 autopsy samples showed AMs positive for SARS-CoV-2 and single-cell transcriptomics revealed nonproductive virus uptake, and activation of inflammatory and anti-viral pathways. Another study which analyzed pulmonary cells from COVID-19 patients found that myeloid cell C-type lectin engagement induced a robust proinflammatory responses that correlated with COVID-19 severity (*Lu et al., 2021*). The robust Spike binding to AMs and their potential roles in COVID-19 lung infection warrants a further exploration of their Spike protein binding partners.

The intranasal administration of the Spike protein resulted in an increase in lung vascular permeability and local tissue damage. While the mechanisms are unknown, infectious agents that target AMs can trigger the secretion of factors that disrupt the integrity of the pulmonary microvascular cell barrier. For example, Porcine reproductive and respiratory syndrome virus-infected AMs trigger transcriptional changes in genes in co-cultured microvascular endothelial cells that reduce vascular integrity (*Sun et al., 2022*). Lung endothelial cells may also be directly targeted (*Bernard et al., 2020*). Spike protein/ACE2 interactions reduce endothelial cell ACE2 expression, which can alter vascular permeability. Yet, the low affinity of mouse ACE2 for the SARS-CoV-2 Spike protein presumably precludes this mechanism in this study. Also arguing against a role for ACE2 in our model, the responses to intranasal Spike protein in hACE2 transgenic mice resembled those of wild-type mice. The Spike protein RGD (arginine-glycine-aspartic acid) motif can also bind $\alpha_v\beta_3$ integrins present on endothelial cells (*Nader and Kerrigan, 2022*). This affects VE-cadherin function and vascular integrity. However, we did not detect any Spike protein on the pulmonary blood vessels following intranasal administration although a low level of binding may have escaped visualization. In sum, the rapid and intense uptake of Spike

protein by AMs supports a role for these cells in the localized neutrophil recruitment, nearby tissue damage, and increased vascular permeability that follows its intranasal administration.

The intravenous administration of the Spike protein led to its accumulation by Kupffer cells in the mouse liver. This uptake was accompanied by a transitory increase in liver sinusoid neutrophils. Human Kupffer cells express LSECtin (liver Siglec and lymph node sinusoidal endothelial cell C-type lectin, CLEC4G) a C-type lectin receptor encoded within the L-SIGN/DC-SIGN/CD23 gene cluster. LSECtin acts as a pathogen attachment factor for Ebola virus and the SARS coronaviruses suggesting that it contributed to the Kupffer cell uptake we noted (*Domínguez-Soto et al., 2009*). The liver sinusoid endothelial cells also avidly accumulated the Spike protein. These cells express another C-type lectin receptor L-SIGN (CD209L/CLEC4M), which has been shown to interact in a $Ca^{2+}$-dependent manner with high mannose-type N-glycans on the SARS-CoV-2 Spike protein (*Kondo et al., 2021*). Based on our mouse studies, blood-borne SARS-CoV-2 Spike proteins, viral particles, or exosomes bearing the Spike protein are likely to accumulate in the liver during severe or prolonged COVID-19 infection.

Local injection of the Spike protein in the region of the cremaster muscle recruited low numbers of neutrophils, but co-administration with IL-1β caused severe neutrophil damage. The Spike protein also proved toxic to purified mouse and human neutrophils cultured ex vivo in its presence. Most human neutrophils bound the recombinant Spike protein. The presence of cations in the binding buffer substantially enhanced the binding suggesting that a cation-dependent receptor accounted for a significant portion of the binding. While the high mannose version bound neutrophils less well, in the NETosis assay it produced a similar level of neutrophil cell death. Neutrophils likely employ several different receptors to capture the Spike protein. Human neutrophils express several C-type lectin receptors including CLEC5A, which has been implicated in SARS-CoV-2-triggered neutrophil NETosis (*Sung et al., 2022*). They also express Siglec-5 (*Connolly et al., 2002*), which bound the Spike protein when expressed in HEK293 cells.

We also assessed the binding of various Spike protein preparations to human peripheral blood mononuclear cells (PBMC) and their impact on cytokine production by M0, M1, and M2 human monocyte-derived macrophages. Monocytes, neutrophils, and B cells bound the full-length trimer best, while T and NK cells exhibited little binding. All the cellar subsets bound more of the SARS-CoV-2 Spike protein in the presence of cations arguing for a major role for cation-dependent receptors. The high mannose Spike protein bound less well to all tested cell types and its binding exhibited the highest cation dependence. Among the eight cytokines measured in the supernatants conditioned by human monocyte-derived macrophages treated with Spike proteins, only TNF-α and IL-6 were elevated compared to untreated or HKU1 S1-treated cells. We found little or no change in IL-1β levels suggesting that the tested Spike protein preparations did not activate inflammasomes. Nor did we find any material difference between recombinant Spike protein prepared from HEK293F and CHO cells. Further studies are needed to identify the monocyte, neutrophil, and B cell receptors that account for the Spike protein binding and additional functional studies to assess the consequences of that binding.

Several conclusions can be drawn from this study. First, instilled SARS-CoV-2 Spike protein and VLPs bearing the Spike protein rapidly accumulated on mouse AMs suggesting a similar uptake by human AMs during an active SARS-CoV-2 infection. In mice, the nasal instillation of the Spike protein is accompanied by lung leukocyte infiltration. Second, murine AMs likely use Siglec-F as a capturing receptor although other receptors contribute. Third, following nasal instillation of the Spike protein, the integrity of the pulmonary vasculature weakens. Fourth, resident and recruited neutrophils suffer, either directly targeted by the Spike protein or secondarily due to the inflammatory response. A similar scenario would be expected during SARS-CoV-2 infection, following direct instillation of the Spike protein in the upper airway of humans, or following an intranasal vaccine that directs the synthesis of the SARS-CoV-2 Spike protein or fragments. Fifth, blood monocytes, B cells, neutrophils, and dendritic cells efficiently bind the Spike protein, while only a low percentage of CD8 T cells and even fewer CD4 T cells do. Spike protein binding to human monocyte-derived macrophages increases the TNF-α and IL-6 secretion. Sixth, altering the glycan composition of the Spike and the D614G mutation had surprisingly little impact on the initial inflammatory response. Seventh, viral particles and

soluble Spike protein that spills into the blood during SARS-CoV-2 infection or following immunization will likely be cleared by liver Kupffer cells but will also target blood vessel endothelial cells in multiple organs. Finally, injection of the Spike protein as occurs following vaccination rapidly delivers it to the draining lymph node via afferent lymphatics. Surprisingly, lymph node medullary macrophages, which largely serve a degradative function, preferentially uptake the Spike protein. Designing a less toxic Spike protein immunogen that targets subcapsular sinus macrophages might better deliver the immunogen to B cells, thereby eliciting a more efficacious antibody response.

# Materials and methods

**Key resources table**

| Reagent type (species) or resource | Designation | Source or reference | Identifiers | Additional information |
|---|---|---|---|---|
| Cell line (hamster, Chinese) | FreeStyle CHO-S Cells | Thermo Fisher Scientific | R80007 | |
| Cell line (Homo sapiens) | FreeStyle 293-F Cells | Thermo Fisher Scientific | R79007 | |
| Cell line (Homo sapiens) | 293T | ATCC | CRL-3216 | |
| Cell line (Homo sapiens) | 293 [HEK-293] | ATCC | CRL-1573 | |
| Cell line (Homo sapiens) | A549 | ATCC | CRM-CCL-185 | |
| Antibody | PerCP-Cy5.5 Anti-Mouse Ly6G (rat monoclonal) | BD Biosciences | RRID: AB_1727563 | FACS (1:800) |
| Antibody | PE Anti-Mouse I-A/I-E (rat monoclonal) | BD Biosciences | RRID: AB_396546 | FACS (1:300) |
| Antibody | PE/Cyanine7 anti-mouse CD24 (rat monoclonal) | BioLegend | RRID: AB_756048 | FACS (1:300) |
| Antibody | PerCP/Cy5.5 anti-mouse CD24 (rat monoclonal) | BioLegend | RRID: AB_1595491 | FACS (1:500) |
| Antibody | Alexa Fluor 647 anti-mouse CD64 (FcγRI) (rat monoclonal) | BioLegend | RRID: AB_2566561 | FACS (1:500) |
| Antibody | Alexa Fluor 488 anti-mouse CD31 (rat monoclonal) | BioLegend | RRID: AB_493408 | FACS (1:300) Intravital (10–20 µl per mouse, i.v.) |
| Antibody | APC-Cy7 Rat Anti-Mouse Siglec-F | BD Biosciences | RRID: AB_2732831 | FACS (1:300) |
| Antibody | PE anti-mouse CD170 (Siglec-F) (rat monoclonal) | BD Biosciences | RRID: AB_394341 | FACS (1:300) IF (1:200) |
| Antibody | Alexa Fluor 647 Anti-Mouse Siglec-F (rat monoclonal) | BD Biosciences | RRID: AB_2687570 | FACS (1:300) |
| Antibody | BV421Anti-Mouse Ly6C (rat monoclonal) | BD Biosciences | RRID: AB_2737748 | FACS (1:500) |
| Antibody | PE/Dazzle 594 anti-mouse CD19 (rat monoclonal) | BioLegend | RRID: AB_2564001 | FACS (1:300) |
| Antibody | Brilliant Violet 421 anti-mouse F4/80 (rat monoclonal) | BioLegend | RRID: AB_2563102 | FACS (1:500) |
| Antibody | PE anti-mouse CD169 (Siglec-1) (rat monoclonal) | BioLegend | RRID:AB_10915697 | FACS (1:500) |
| Antibody | PE anti-human CD16 (mouse monoclonal) | BioLegend | RRID: AB_2562749 | FACS (1:800) |
| Antibody | PE/Cyanine7 anti-human CD56 (NCAM) (mouse monoclonal) | BioLegend | RRID: AB_2563927 | FACS (1:600) |
| Antibody | APC Anti-Human CD4 (mouse monoclonal) | BD Bioscience | RRID: AB_398521 | FACS (1:500) |
| Antibody | APC/Cyanine7 anti-human CD8a Antibody | BioLegend | RRID: AB_314134 | FACS (1:500) |
| Antibody | Brilliant Violet 421 anti-human HLA-DR (mouse monoclonal) | BioLegend | RRID: AB_2561831 | FACS (1:1000) |
| Antibody | Brilliant Violet 650 anti-human CD14 (mouse monoclonal) | BioLegend | RRID: AB_2563799 | FACS (1:500) |
| Antibody | Brilliant Violet 711 anti-human CD20 Antibody | BioLegend | RRID: AB_2562602 | FACS (1:300) |

*Continued on next page*

*Continued*

| Reagent type (species) or resource | Designation | Source or reference | Identifiers | Additional information |
|---|---|---|---|---|
| Antibody | PE/Cyanine7 anti-human CD20 (mouse monoclonal) | BioLegend | RRID: AB_314260 | FACS (1:300) |
| Antibody | PE anti-human CD123 (mouse monoclonal) | BioLegend | RRID: AB_314580 | FACS (1:100) |
| Antibody | APC/Cyanine7 anti-human CD15 (SSEA-1) (mouse monoclonal) | BioLegend | RRID: AB_2750190 | FACS (1:400) |
| Antibody | APC anti-human CD66b (mouse monoclonal) | BioLegend | RRID: AB_2566607 | FACS (1:500) |
| Antibody | Brilliant Violet 421 anti-human CD11c (mouse monoclonal) | BioLegend | RRID: AB_2564485 | FACS (1:200) |
| Antibody | BD Pharmingen PerCP-Cy5.5 Anti-Human CD3 (mouse monoclonal) | BDBioscience | RRID: AB_394493 | FACS (1:300) |
| Antibody | PE Anti-Human CD22 (mouse monoclonal) | BD Bioscience | RRID: AB_2737845 | FACS (1:100) |
| Antibody | APC anti-human CD170 (Siglec-5) (mouse monoclonal) | BioLegend | RRID: AB_2564262 | FACS (1:200) |
| Antibody | APC anti-human Siglec-8 (mouse monoclonal) | BioLegend | RRID: AB_2561402 | FACS (1:200) |
| Antibody | Alexa Fluor 647 anti-mouse CD169 (Siglec-1) (rat monoclonal) | BioLegend | RRID: AB_2563621 | FACS (1:300) IF (1:500) |
| Antibody | Human Siglec-8 (mouse monoclonal) | R&D Systems | MAB7975 | FACS (1:200) |
| Antibody | PE-Human Siglec-8 (mouse monoclonal) | R&D Systems | RRID: AB_2905537 | FACS (1:200) |
| Antibody | Alexa Fluor 647-Human Siglec-1/CD169 (mouse monoclonal) | R&D Systems | RRID: AB_2905550 | FACS (1:300) IF (1:200) |
| Antibody | Alexa Fluor 647- Human ACE-2 (mouse monoclonal) | R&D Systems | FAB9332R100UG | FACS (1:300) |
| Antibody | Alexa Fluor 750-Human Siglec-8 (mouse monoclonal) | R&D Systems | FAB7975S- 100UG | FACS (1:300) |
| Antibody | Alexa Fluor 405-Human ACE-2 (mouse monoclonal) | R&D Systems | FAB9332V- 100UG | FACS (1:300) |
| Antibody | Human/Mouse/Rat/Hamster ACE-2 Antibody (goat polyclonal) | R&D Systems | RRID: AB_355722 | FACS (1:300) Blocking (0.5 µg per test) |
| Antibody | SARS-CoV-2 (2019-nCoV) Spike Neutralizing Antibody (rabbit monoclonal) | SinoBiological | RRID: AB_2857936 | FACS (1:300) Blocking (0.5 µg per test) |
| Antibody | Rat IgG derivative against the GPIbβ subunit of the murine platelet/megakaryocyte-specific GPIb-V- IX complex antibody (rat IgG derivative) | Emfret | RRID: AB_2861336 | Intravital imaging (2 µl per mouse, i.v.) |
| Antibody | Mouse LYVE-1 Antibody (rat monoclonal) | R&D Systems | RRID: AB_2138528 | IF (1:400) |
| Antibody | PE anti-mouse Podoplanin Antibody (Syrian Hamster monoclonal) | BioLegend | RRID: AB_2161928 | IF (1:200) |
| Commercial assay or kit | LIVE/DEAD Fixable Aqua Dead Cell Stain Kit | Thermo Fisher Scientific | L34966 | FACS (1:1000) |
| Commercial assay or kit | Lenti-X Concentrator | Takara Bio USA Inc | 631232 | |
| Chemical compound, drug | Evans blue | Sigma-Aldrich | E2129-10G | |
| Chemical compound, drug | Propidium iodide (PI) | Sigma-Aldrich | P4864-10ML | |
| Chemical compound, drug | FluoSpheres NeutrAvidin-Labeled Microspheres, 0.2 µm, yellow-green fluorescent (505/515), 1% solids | Thermo Fisher Scientific Inc | F8774 | |
| Recombinant DNA reagent | SIGLEC5 (NM_003830) Human Tagged ORF Clone | Origene | RC206610 | |

*Continued on next page*

*Continued*

| Reagent type (species) or resource | Designation | Source or reference | Identifiers | Additional information |
|---|---|---|---|---|
| Recombinant DNA reagent | SIGLEC5 (NM_003830) Human Tagged ORF Clone | Origene | RG206610 | |
| Recombinant DNA reagent | Human Siglec-8 (NP_055257) VersaClone cDNA | Origene | RDC1496 | |
| Recombinant DNA reagent | SARS-CoV-2 Spike-S | Addgene | Plasmid # 154754 | *Hsieh et al., 2020* |
| Recombinant DNA reagent | HIV-1 NL4-3 Gag-iGFP ΔEnv | NIH AIDS Reagent Program | 12455 | |
| Recombinant DNA reagent | Human Coronavirus Spike glycoprotein Gene ORF cDNA clone expression plasmid (Codon Optimized) HCoV-HKU1 | SinoBiological | VG40021-UT | |
| Recombinant DNA reagent | Human coronavirus (HCoV-229E) Spike Gene ORF cDNA clone expression plasmid (Codon Optimized) HCoV-229E | SinoBiological | VG40605-UT | |
| Recombinant DNA reagent | ACE2 cDNA ORF Clone, Human, C-OFPSpark tag | SinoBiological | HG10108-ACR | |
| Recombinant DNA reagent | pCMV-dR8.2 dvpr | Addgene | 8455 | |
| Recombinant DNA reagent | pLentipuro3 TO V5-GW EGFP-Firefly Luciferase | Addgene | 119816 | |
| Peptide, recombinant protein | ACE2 Protein, Human, Recombinant (mFc Tag) | SinoBiological | 10108-H05H | |
| Peptide, recombinant protein | Human coronavirus HKU1 (isolate N5) (HCoV-HKU1) Spike/S1 Protein (S1 Subunit, His Tag) | SinoBiological | 40602-V08H | |
| Peptide, recombinant protein | Human coronavirus (HCoV-229E) Spike Protein (S1+S2 ECD, His Tag) | SinoBiological | 40605-V08B | |
| Peptide, recombinant protein | SIGLEC5 Protein, Human, Recombinant (hFc Tag) | SinoBiological | 11798-H02H | |
| Peptide, recombinant protein | Recombinant Mouse Siglec-F Fc Chimera Protein, CF | R&D Systems | 1706-SF-050 | |
| Peptide, recombinant protein | Recombinant Human Siglec-8 Fc Chimera Protein, CF | R&D Systems | 9045-SL-050 | |
| Strain, strain background (mouse) | C57BL/6J | Jackson Lab. | IMSR_JAX:000664 | Jax stock 000664 |
| Strain, strain background (mouse) | K18-hACE2 (B6.Cg- Tg(K18- ACE2) 2Prlmn/J) | Jackson Lab. | IMSR_JAX:034860 | Jax stock 034860 |
| Other | Liberase TL(Thermolysin Low) Research Grade | Roche Applied Science | 5401020001 | Enzyme |
| Other | Opti-MEM | Thermo Fisher Scientific Inc. | 31985070 | Cell culture media |
| Other | RPMI 1640 Media | Thermo Fisher Scientific Inc. | 11875093 | Cell culture media |
| Other | TransIT-293 Transfection Reagent | Mirus Bio LLC | MIR 2704 | Cell culture media |
| Other | FreeStyle 293 Expression Medium | Thermo Fisher Scientific Inc. | 12338018 | Cell culture media |
| Other | FreeStyle CHO Expression Medium | Thermo Fisher Scientific Inc. | 12651014 | Cell culture media |

## Mice

C57BL/6 and K18-ACE2 transgenic mice were obtained from Jackson Laboratory. All mice (female) used in this study were 8–12 weeks of age. Mice were housed under specific pathogen-free conditions.

All the animal experiments and protocols used in the study were approved by the National Institute of Allergy and Infectious Diseases (NIAID) Animal Care and Use Committee (ACUC) at the National Institutes of Health.

## Cells

To isolate mouse lung cells, lungs were carefully collected and gently teased apart using forceps into RPMI 1640 media containing 2 mM L-glutamine, antibiotics (100 IU/ml penicillin, 100 µg/ml streptomycin), 1 mM sodium pyruvate, and 50 µM 2-mercaptoethanol, pH 7.2. The tissue was then digested with Liberase Blendzyme 2 (0.2 mg/ml, Roche Applied Science) and DNase I (20 µg/ml) for 1 hr at 37°C. The proteases were inactivated by adding 10% fetal bovine serum and 2 mM EDTA and the cell disaggregated by passing them through a 40 µm nylon sieve (BD Bioscience). Single cells were then washed with 1% bovine serum albumin (BSA) in PBS and blocked with anti-Fcγ receptor (BD Biosciences). Human PBMC were purified from whole blood by density gradient centrifugation (FicollPaque, Miltenyi Biotec). Whole blood was collected from healthy donors through an NIH Department of Transfusion Medicine (DTM)-approved protocol (Institutional Review Board of the NIAID). Neutrophil and monocyte cell population were each obtained by negative selection (>97% purity, Stem Cell Technologies). To generate human monocyte-derived macrophages, purified monocytes were treated for 7 days with 50 ng/ml human recombinant M-CSF (PeproTech) in RPMI medium supplemented with 10% heat-inactivated fetal bovine serum, and 1 mM sodium pyruvate, 100U Penicillin-Streptomycin (Gibco, Thermo Fisher Scientific) in ultra-low attachment culture 100 mm dishes. On day 7 the mature macrophages were collected and verified to be more than 90% CD68$^+$ by flow cytometry.

## Reagents

See Key resources table.

## Flow cytometry

Single cells were re-suspended in 0.1% fatty acid-free BSA-HBSS with 100 µM CaCl$_2$ and 1 mM MnCl$_2$ unless otherwise specified. Buffer without divalent cations included 10 mM EDTA. The cells were stained with fluorochrome-conjugated antibodies against various cell surface markers or with different fluorochrome-conjugated proteins, which are listed in the resource and reagent tables in the supplement. LIVE/DEAD Fixable Aqua Dead Cell Stain Kit, LIVE/DEAD Fixable NearIR Dead Cell Stain Kit, or LIVE/DEAD Fixable Yellow Dead Cell Stain Kit (Thermo Fisher) were used in all experiments to exclude dead cells. Compensation was performed using AbC Total Antibody Compensation Bead Kit (Thermo Fisher) and ArCTM Amine Reactive Compensation Bead (Thermo Fisher) individually stained with each fluorochrome. Compensation matrices were calculated with FACSdiva software. Data acquisition including cell number count was done on a FACSCelesta SORP (BD) flow cytometer and analyzed with FlowJo 10.8.x software (Tree Star).

## Human macrophage polarization and cytokine profiling

To polarize human macrophages, monocyte-derived macrophages were plated in 48-well plates at 5×10$^4$ cells per well in 500 µl RPMI medium and allowed to rest for 2 hr at 37°C before treating cytokines or LPS. The human M0 macrophages were either left untreated or treated for 48 hr to induce macrophage polarization: (i) for M1 polarization with 10 ng/ml LPS (E055:B55; Sigma-Aldrich) and 20 ng/ml IFNγ (PeproTech), (ii) for M2 polarization with 50 ng/ml human recombinant M-CSF, 20 ng/ml human recombinant IL-4, and 20 ng/ml human recombinant IL-13 (PeproTech). Human macrophages were then cultured alone or with 0.1 or 1 µg SARS-CoV-2 Spike proteins for 48 hr. The cultured supernatants were collected, and cytokine levels determined using the bead-based immunoassays LEGENDplex Human Macrophage/Microglia Panel (BioLegend). The assays were performed in 96-well plates following the manufacturer's instructions. For measurements, a FACSCelesta SORP flow cytometer (BD Biosciences) was employed, and data were evaluated with the LEGENDplex Data Analysis software.

## Production and purification of recombinant SARS-CoV-2 Spike proteins (see Figure 1—figure supplement 1)

SARS-CoV-2 Spike expression vectors were purchased from Addgene (Key resources table). Endotoxin-free plasmids were prepared by Alta Biotech and provided at a concentration of 5 mg/ml. Transfections were carried out using CHO Freestyle cells (Invitrogen). 1.24 mg of plasmid DNA was transfected into 90 million cells, using a MaxCyte Electroporation Transfection System. A detailed protocol is available at https://maxcyte.com/atx. Culture supernatants were harvested on day 6 and clarified by centrifugation, followed by filtration through a 0.45 µm filter, followed by the addition of EDTA-free protease inhibitor cocktail tablets (Roche). Culture supernatants were then dialyzed overnight at 4°C in HBS (150 mM NaCl, 10 mM HEPES, pH 8.0) using 10 kDa MWCO Slide-A-Lyzer dialysis cassettes (Thermo Scientific). Supernatants were passed over a 10 ml StrepTrap HP column (Cytiva) at 1 ml/min using an ÄKTA pure 150 purifier (Cytiva), maintained at 4°C. Bound protein was eluted with elution buffer (2.5 mM desthiobiotin, 100 mM Tris-Cl, 150 mM NaCl, 1 mM EDTA, pH 8.0). Peak fractions were pooled and concentrated using 30 kDa MW CO Amicon centrifugal concentrators (Millipore). Trace endotoxins were removed by two sequential Triton X-114 extractions, followed by passage through an HiPPR detergent removal column (Thermo Fisher). Following the removal of endotoxin, the remaining contamination was assessed using the Pierce Chromogenic Endotoxin Quant Kit (Thermo Fisher), based on the amebocyte lysate assay. The endotoxin level in the purified recombinant protein preparation is below 1.0 EU/ml, which closely aligns with the levels specified by the company for recombinant proteins. Protein concentrations were determined by a BCA Protein assay (Thermo Fisher).

## VLPs and lentivirus preparation

SARS-CoV-2 Spike protein-incorporated NL4.3-GFP VLPs were produced by transfecting HEK293T cells with full-length SARS-CoV-2 Spike-S and HIV-1 NL4-3 Gag-iGFP ΔEnv (12455, NIH AIDS Reagent Program) at a ratio of 1:2.5 using a previously reported method (*Park et al., 2015*). EGFP control lentiviruses (no Spike protein) were produced by transfecting HEK293T cells with pCMV-dR8.2 dvpr (packaging) and pLentipuro3 TO V5-GW EGFP-Firefly Luciferase (reporter). Coronavirus Spike proteins were incorporated by co-transfection with SARS-CoV-2 Spike-S, HCoV-229E Spike, or HCoV-HKU1 Spike expressing plasmids. Eighty percent confluent HEK293T in six-well plates (2.5 ml/well) were transfected with 5 µg of plasmid diluted in Opti-MEM with a 1:4 (DNA/reagent) dilution of TransIT-293 Transfection Reagent. The media was harvested 64 hr later and centrifuged at 500×$g$ for 10 min at 4°C. The supernatants were collected and mixed with Lenti-X Concentrator (Takara Bio USA, Inc) at a 1:3 ratio. The mixture was placed at 4°C overnight and centrifuged the following day at 1500×$g$ for 45 min. Fluorescent VLPs were directly counted and measured by BD FACSCelesta flow cytometer. The forward scatter (FSC) detector (photodiode with 488/10 BP filter) and side scatter (SSC) detector (photomultiplier tube [PMT] with 488/10 BP filter) were tuned to detect voltages up to 530 for FSC and 220 for SSC. The NL4.3-GFP VLPs were distinguished from noise and non-VLP particles by GFP signals. NL4.3-GFP VLP Spike protein incorporation was tested using hACE2 expressing HEK293 cells (ACE2/HEK293). Ten thousand VLPs were incubated with 1×10$^4$ ACE2/HEK293 cells in 1× HBSS (contains 1 mM Ca$^{2+}$, 2 mM Mg$^{2+}$, 1 mM Mn$^{2+}$, and 0.5% fatty acid-free BSA) buffer at 4°C for 30 min. Binding of the VLPs to ACE2/HEK293 cells was measured with BD FACSCelesta. The 50% lentivirus transduction dose was determined using percentile (50%) of GFP-positive ACE2/HEK293 cells 3 days after transduction. The pelleted VLPs or lentiviruses were suspended in PBS and frozen in single use aliquots.

## Lentivirus transduction assay

96-well plates were seeded with 200 µl of ACE2/HEK293 cells (1.5×10$^4$ cells/ml). After a 24 hr incubation, recombinant proteins or neutralizing antibody was added. Lentiviral particles capable of transducing 50% of the cells (CoV2-lenti, HKU1-lenti, and 229E-lenti) were added (~25 µl) 30 min later. Following a 3-day culture, the ACE2/HEK293 cells were harvested and GFP expression levels determined by BD FACSCelesta flow cytometry. Each group contained two to five replicates.

## Measurement of vascular permeability

Pulmonary vascular permeability was measured by i.v. administration of Evans blue dye (0.2 ml 0.5% in PBS) (*Zhou et al., 2011*). One and half hours after Spike protein nasal administration, Evans blue

dye was injected intravenously. After another 1.5 hr the mice were perfused with PBS, and lungs and livers were harvested, and dye extracted in formamide overnight at 55°C. Dye concentrations were quantified by measuring absorbance at 610 nm with subtraction of reference absorbance at 450 nm. The content of Evans blue dye was determined by generating a standard curve from dye dilutions.

## NETosis assays

Mouse bone marrow-derived or human neutrophils (>97% purity) were re-suspended in RPMI 1640 media ($1\times10^6$/ml) and incubated at 37°C in 5% $CO_2$ for 30 min. To induce NETosis, the cells were exposed to fMLP (1 μM); LPS (10 ng/ml); TNF-α (20 ng/ml); PMA (30 nM), or four different SARS-CoV Spike protein preparations (0.1 or 1 μg/ml) for 4 hr to overnight at 37°. The cultures were terminated by adding 4% PFA for 15 min. Helix NP NIR (0.1 μM; BioLegend) and DAPI (0.3 nM; BioLegend) were added to detect NETs. Data acquisition (Helix NP NIR[+] DAPI[+] Cells) was done on FACSCelesta SORP (BD) flow cytometer and analyzed with FlowJo software (Tree Star). In vitro imaging of NETosis was performed using a modified protocol from a previous report (*Hoppenbrouwers et al., 2017*). The A549 cells were plated at 48 hr before imaging at 60% cell confluent. Fluorescently labeled SARS-CoV-2 Spike protein (Alexa Fluor 488; 0.5 μg/ml) was added to the A549 cells 24 hr before human neutrophil seeding. Human neutrophils were purified from whole blood by EasySep Direct Human Neutrophil Isolation Kit (Stem Cell Technologies). Purified human neutrophils were stained with Hoechst prior to seeding on SARS-CoV-2 Spike protein-pretreated A549 cells in propidium iodide (PI, 1 μg/ml) containing culture media. Time-lapse images were acquired with a Leica SP8-inverted five-channel confocal microscope (Leica Microsystems) equipped with 40× oil objective, 0.95 NA (immersion medium used distilled water). The temperature of air (5% $CO_2$) was maintained at 37.0 ± 0.5°C. Time interval between frames was set as 10 min for 5 hr acquisition. The loss of nucleus of human neutrophils was detected by the loss of Hoechst signal and extracellular DNA released by NETosis was labeled by PI signals.

## Fluorescent nanobead binding assay of SARS-CoV-2 Spike protein

FluoSpheres NeutrAvidin-Labeled Microspheres, 0.2 μm, yellow-green fluorescent (505/515), 1% solids (Cat# F8774, Thermo Fisher Scientific Inc) was used as a nanobead platform for SARS-CoV-2 Spike protein conjugation. Fluorescent nanobead was directly counted and measured by BD FACSCelesta flow cytometer. A flow cytometer equipped with FSC detector (Photodiode with 488/10 BP filter) and SSC detector (PMT with 488/10 BP filter) was tuned to detect voltages up to 550 for FSC and 230 for SSC. Million counts of nanobeads were conjugated with 0.5 μg of SARS-CoV-2 Spike protein by interaction of Neutravidin on beads and Strep Tag II on recombinant proteins. Nanobeads and SARS-CoV-2 Spike proteins in PBS were coupled at room temperature for 1 hr and washed with 1 ml of PBS. The coupled nanobeads were spun down with a benchtop centrifuge at a speed of 20,000 ×g, at 4°C for 20 min. The nanobeads' pellet was suspended with PBS concentration at $2\times10^4$/μl. SARS-CoV-2 Spike protein couplings to nanobeads were tested with Alexa Fluor 647-conjugated SARSCoV-2 Spike neutralizing antibody (Cat# 40591-MM45, SinoBiological). To test the binding ability of various recombinant proteins (hACE2, human Siglec-5, human Siglec-8, mouse ACE2, and mouse Siglec-F) were directly conjugated with Alexa Fluor 647. SARS-CoV-2 Spike protein-conjugated nanobeads ($2\times10^4$ counts) in 20 μl of 1× HBSS (contains 1 mM $Ca^{2+}$, 2 mM $Mg^{2+}$, 1 mM $Mn^{2+}$, and 0.5% fatty acid-free BSA) buffer were incubated with 0.2 μg of fluorescent recombinant proteins at room temperature for 30 min. Fluorescent antibody or recombinant protein biding on fluorescent nanobeads was directly counted and measured by BD FACSCelesta flow cytometer.

## Thick section immunohistochemistry and confocal microscopy

Immunohistochemistry was performed using a modified method of a previously published protocol (*Park et al., 2015*; *Park et al., 2018*). Briefly, freshly isolated lungs were fixed in newly prepared 4% paraformaldehyde (Electron Microscopy Science) overnight at 4°C on an agitation stage. Fixed lungs were embedded in 4% low melting agarose (Thermo Fisher Scientific) in PBS and sectioned with a vibratome (Leica VT-1000 S) at a 50 μm thickness. Thick sections were blocked in PBS containing 10% fetal calf serum, 1 mg/ml anti-Fcγ receptor (BD Biosciences), and 0.1% Triton X-100 (Sigma) for 30 min at room temperature. Sections were stained overnight at 4°C on an agitation stage with Ly6G, Siglec-F, CD169, CD31, and F4/80, and labeled WGA. Stained thick sections were microscopically

analyzed using a Leica SP8 confocal microscope equipped with an HC PL APO CS2 40× (NA, 1.30) oil objective (Leica Microsystem, Inc) and images were processed with Leica LAS AF software (Leica Microsystem, Inc) and Imaris software v.9.9.1 64× (Oxford Instruments plc). The intensities of fluorescent signals in regions of interests were measured by LSA AF Lite software (Leica Microsystem).

### In vitro imaging of HEK293 cells

HEK293 cells were transfected using TransIT-293 transfection reagent (Mirus Bio LLC). For transient expression of human Siglec-5-GFP, 80% confluent HEK293 in eight-well chamber slide (0.25 ml/well) were transfected by adding dropwise to each well 25 µl containing 0.25 µg of plasmid diluted in Opti-MEM with a 1:4 (DNA/reagent) dilution of TransIT-293 Transfection Reagent. To generate a stable cell line of human Siglec-5-GFP and hACE2-OFP, transfected HEK293 cells were sorted with FACS Aria II by GFP and OFP signals. Sorted human Siglec-5-GFP transfected HEK293 cells were further selected with G418 (0.8 mg/ml) containing growth media and maintained with G418 (0.5 mg/ml) media. Sorted hACE2OFP transfected HEK293 cells were further selected with hygromycin B (0.2 mg/ml) containing growth media and maintained with hygromycin B (0.1 mg/ml) media. Transiently transfected HEK293 cells in eight-well chamber slide were directly imaged with a Leica SP8-inverted five-channel confocal microscope (Leica Microsystems). HEK293 cells, HEK293 Siglec-5-GFP, and HEK293 ACE2-OFP cell lines were plated at a ratio of 1:1:1 (cell numbers) in eight-well chamber with normal media 18 hr prior to imaging. Imaging was performed with a confocal microscope equipped with 40× oil objective, 0.95 NA (immersion medium used distilled water). The temperature of air (5% $CO_2$) was maintained at 37.0 ± 0.5°C. Fluorescent SARS-CoV-2 Spike protein (Alexa Fluor 647) (1 µg/ml) was added into culture media. Images were acquired with Leica LAS AF software (Leica Microsystem, Inc) and processed with Imaris software v.9.9.1 64× (Oxford Instruments plc).

### Time-lapse imaging of lung sections with confocal microscopy

Lung slices were obtained from mouse lungs using a slightly modified published protocol (*Pieretti et al., 2014*). Briefly, C57BL/6 mice were euthanized with overdose of Avertin (1 ml of 2.5% Avertin). The peritoneum was opened, and the descending aorta cut allowing blood to pool in the abdomen. The trachea was cannulated, and the lungs inflated with 37°C 1.5% low-melting-point agarose (Cat# 50111, Lonza) prepared with RPMI 1640 media. Subsequently, the lungs were excised and rinsed with RPMI 1640 media. Isolated left lungs were sectioned into six to eight 1-mm-thick transverse slices using a #10 scalpel blade. Lung slices were placed in a pre-warmed cover glass chamber slide (Nalgene, Nunc) under a metal flat washer (M8-5/16th inches diameter). The chamber slide was then placed into the temperature control chamber on the microscope. The temperature of air was monitored and maintained at 37.0 ± 0.5°C for 5% $CO_2$. Mounted lung sections on the chamber slide were microscopically analyzed using a Leica SP8 confocal microscope equipped with an HC PL APO CS2 40× (NA, 1.30) oil objective (Leica Microsystem, Inc) and images were processed with Leica LAS AF software (Leica Microsystem, Inc). Lung slices were imaged a range of depths (10–50 µm). Time-lapse images were processed with Imaris software v.9.9.1 64× (Oxford Instruments plc).

### Intravital imaging

The microanatomy of liver, spleen, heart, Peyer's patch, and inguinal LN were delineated by tail vein injection of labeled antibodies before imaging. Antibodies used included CD31, blood vessels; F4/80, Kupffer cells and macrophages; CD169, subcapsular macrophages; and Ly6G; neutrophils. The antibody mixtures were injected 10 min before starting animal surgery. Fluorescently labeled Spike proteins were injected intravenously or at the mouse tail base as indicated. To image the liver or spleen a slightly modified published protocol was used (*Matsumoto et al., 2018*). Briefly, after initial anesthesia (Avertin 300 mg/kg, i.p.) the skin and peritoneum were cut to expose the left lobe of the liver, or the left flank was cut below the costal margin to expose the spleen. The visible organs were glued with *n*-butyl cyanoacrylate to a custom-made metal holder. After attachment, the mouse was placed over a pre-warmed cover glass (Brain Research Laboratories) window on universal mounting frame AK-Set (PECON). The exposed organs were kept moist with saline wetted gauze. The mounting frame was placed into the temperature control chamber on the microscope and maintained at 37.0 ± 0.5°C. Once stabilized onto imaging stage/insert the mice received isoflurane (Baxter; 2% for induction of anesthesia, and 1–1.5% for maintenance, vaporized in an 80:20 mixture of oxygen and air). For

liver and spleen four-dimensional analysis of cell behavior, stacks of various numbers of section (z-step = 3, 5) were acquired every 5–30 s to provide an imaging volume of 20–50 μm in depth. Intact organ imaging of the heart, inguinal lymph node, or Peyer's patches was performed after mouse sacrifice using same imaging procedure as used for liver/spleen imaging.

To image neutrophils in the cremaster muscle (*Yan et al., 2021*), mice received an intrascrotal injection of PBS, BSA, IL-1β (50 ng in 300 μl saline, R&D Systems), Spike proteins, or IL-1b and Spike proteins. 90 min prior to imaging, the mice received injections of Avertin (300 mg/kg, i.p.) and fluorescently labeled antibodies intravenously. Antibodies used directed against Gr-1, neutrophils; GPIbβ, platelets; and CD31, blood vessel endothelium. The isolated cremaster tissue was exteriorized and stabilized onto the imaging stage/insert with the tissue directly contacting the cover glass. The exposed tissue was kept moist with pre-warmed saline (37°C). Once stabilized onto an imaging stage/insert the mouse received isoflurane (Baxter; 2% for induction of anesthesia, and 1–1.5% for maintenance, vaporized in an 80:20 mixture of oxygen and air), and placed into a temperature-controlled chamber. Image stacks of optical sections (z-step = 1) were routinely acquired at 20–30 s intervals to provide an imaging volume of 15–25 μm. All imaging was performed with a Leica SP8-inverted five-channel confocal microscope (Leica Microsystems) equipped with HC PL APO CS2 40× (NA, 1.30) oil objective. Sequences of image stacks were transformed into volume-rendered four-dimensional videos using Imaris software v.9.9.1 64× (Oxford Instruments plc). Video editing was performed using Adobe Premiere Pro 2022 (Adobe Systems Incorporated).

## Quantifications and statistical analyses

All experiments were performed at least three times. Representative images were placed in figures. Primary image data which was analyzed and calculated by Leica LAS AF software (Leica Microsystem, Inc) or Imaris software v.9.9.1 64× (Oxford Instruments plc). was acquired and processed with Microsoft Excel software. Error bars with ± SEM, and p values were calculated with unpaired t-test or two-way ANOVA multiple comparisons in GraphPad Prism 9.3.1 (GraphPad software). $p < 0.05$ was considered significantly different.

## Acknowledgements

The authors thank Dr. Anthony Fauci for long-standing encouragement. This work was supported by Intramural Research Program of National Institute of Allergy and Infectious Diseases.

## Additional information

### Funding
No external funding was received for this work.

### Author contributions
Chung Park, Conceptualization, Resources, Data curation, Formal analysis, Investigation, Methodology, Writing - review and editing; Il-Young Hwang, Conceptualization, Data curation, Formal analysis, Investigation, Methodology; Serena Li-Sue Yan, Data curation, Formal analysis, Methodology; Sinmanus Vimonpatranon, Don Van Ryk, Alexandre Girard, Resources, Methodology; Danlan Wei, Resources; Claudia Cicala, James Arthos, Conceptualization, Resources, Methodology; John H Kehrl, Conceptualization, Supervision, Funding acquisition, Investigation, Writing - original draft, Project administration, Writing - review and editing

### Author ORCIDs
Chung Park ⬤ http://orcid.org/0000-0002-7819-5333
Il-Young Hwang ⬤ http://orcid.org/0000-0003-4498-4382
John H Kehrl ⬤ http://orcid.org/0000-0002-6526-159X

### Ethics
This study strictly adhered to the recommendations in the Guide for the Care and Use of Laboratory Animals by the National Institutes of Health. All animals were handled according to approved

institutional animal care and use committee (IACUC) protocols (LIR-15) of the National Institutes of Allergy and Infectious Disease. The protocol received approval from the Committee on the Ethics of Animal Experiments of the National Institutes of Allergy and Infectious Disease. All surgeries were performed under anesthesia, with every effort made to minimize suffering.

Reviewer #1 (Public Review): https://doi.org/10.7554/eLife.86764.3.sa1
Reviewer #3 (Public Review): https://doi.org/10.7554/eLife.86764.3.sa2
Author Response https://doi.org/10.7554/eLife.86764.3.sa3

## Additional files

### Supplementary files
• MDAR checklist

### Data availability
Figure 1—figure supplement 2—source data 1, Figure 2—source data 1,2, Figure 2—figure supplement 1—source data 1, Figure 3—source data 1, Figure 3—figure supplement 1—source data 1, Figure 4—figure supplement 1—source data 1, Figure 5—source data 1,2, Figure 7—source data 1–3, and Figure 8—source data 1 contain the numerical data used to generate the figures. Primary imaging files are available as Figure 1—source data 1–6, Figure 1—figure supplement 2—source data 2,3, Figure 3—source data 2, Figure 4—figure supplement 3—source data 1, and Figure 7—source data 4.

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
