## [Editor Report · eLife assessment]

This paper investigates the impact of intranasal instillation of SARS CoV2 spike protein in mouse models of lung inflammation. The authors conclude that the spike protein can interact with macrophages through carbohydrate recognition and can induce recruitment and NETosis of neutrophils, contributing to lung inflammation. They also use the cremaster muscle model to investigate effect of the spike proteins on neutrophil dynamics and death using intravital microscopy. Given that mucosal vaccines using SARS CoV2 spike variants could be envisioned as desirable, the observation that spike can induce lung/mucosal inflammation even without an adjuvant is **important**. Despite limitations of some loose terminology and some weak controls, the key observations are **solid** and demand further attention given the importance of the antigen.

---

## [Referee Report · Reviewer #1 (Public Review)]

The manuscript by Park et. al. examines the interaction of macrophages with SARS-CoV-2 spike protein and subsequent inflammatory reactions. The authors demonstrate that following intranasal delivery of spike it rapidly accumulates in alveolar macrophages. Inflammation associated with internalized spike recruits neutrophils to the lung, where they undergo a cell death process consistent with NETosis. The authors demonstrate that modifications spike to contain high mannose reduces uptake of spike protein and limits the inflammation induced. This finding could have implications for vaccine development, as vaccines containing modified spike could be safer and better tolerated.

The authors use a number of different techniques, including in vivo modeling, imaging, human and murine systems to interrogate their hypotheses. These systems provide robust supporting information for their conclusions. There are two key aspects from the current manuscript which would add key evidence. The authors suggest that neutrophils exposed to spike protein undergo a process of NETosis. To confirm this hypothesis inhibitors of NETosis should be used to demonstrate that the cell death is prevented. Additionally, vaccination of a murine model with the modified spike protein would add additional support to the conclusion that modified spike protein would be less inflammatory while maintaining its utility as a vaccine antigen.

---

## [Referee Report · Reviewer #3 (Public Review)]

The study focuses on in vivo and in vitro cellular responses intranasal instillation of glycoforms and mutants of SARS-CoV2 spike trimer or spike bearing VLP in mice. Collectively, the experiments suggest that SARS-CoV2 spike has pro-inflammatory roles through increase M1 macrophage associated cytokines and induction of neutrophil netosis/necrosis, a proinflammatory cell death pathway. These effects seem largely independent of hACE2 interaction and partly depend upon interactions with SIGLECs on macrophages and neutrophils. A strength of the study is that a number sophisticated methods are used, including intravital microscopy in the cramaster and liver as well as acute lung slice models, to look at uptake of the spike proteins and immune cell dynamics. The weakness is that some of the reagents maybe contaminated with uncharacterized glycoforms and some important controls, such as control spike protein and control VLP are unevenly applied or not included. The authors have revised the manuscript through some improvements in the writing, but the survey nature and suggestive level of evidence is still a weakness. The study calls attention to sources of proinflammatory activity in the SARS CoV2 spike that may involve some carbohydrate interactions.

---

## [Author Response]

The following is the authors’ response to the original reviews.

**Reviewer #1 (Recommendations For The Authors):**
All comments made in the public section.

We would like to thank the reviewer for their assessment of our study and for suggestions for additional experiments to follow up our studies.

**Reviewer #2 (Recommendations For The Authors):**
‐ Preparation of spike proteins and VLPs. Although Triton‐X114 extraction was done to remove endotoxin from the recombinant spike protein preparations, its removal efficiency depends on the levels of endotoxin in the samples. Therefore, the residual endotoxin levels in each of the test samples and batches should be measured. Even very low but varying levels of residual endotoxin would substantially impact the reported results, as they create inconsistent data that are not interpretable.

Certainly, endotoxin contamination in instilled materials is always an issue. Established protocols for inducing acute inflammatory responses using endotoxin outline specific ranges of endotoxin levels in the instillation materials. To induce acute lung inflammation in mice at least 2 µg of endotoxin must be instilled. We have endeavored to reduce the possibility of endotoxin contamination in our recombinant proteins by using a mammalian expression system; careful aseptic culture and protein purification techniques; and a final Triton-X114 partitioning protocol. We assessed the possibility of endotoxin contamination using the Pierce Chromogenic Endotoxin Quant Kit, which is based on the amebocyte lysate assay. Our analysis revealed that the endotoxin level in the purified recombinant protein preparation is below 1.0 EU/ml, which closely aligns with the levels specified for recombinant proteins. An endotoxin concentration of 1.0 EU/ml is equivalent to approximately 0.1 ng/ml. Throughout all mouse nasal instillation experiments, the total volume of recombinant protein administered did not exceed 6 µl. The amount of contaminant endotoxin instilled did not exceed 1 pg (50 µl of 0.02 ng/ml of endotoxin). Consequently, we can confirm that the extent of endotoxin contamination is at trace levels. Moreover, our study reveals multiple results indicating that the level of endotoxin contamination in the recombinant protein was inadequate to independently induce neutrophil recruitment in the cremaster muscle, lymph nodes, and liver. For further insights, refer to Figure 5.

‐ Doses of spike and VLPs: The amount of spike protein incorporated into HIV Gag‐based VLPs should be determined and compared to that found in the native SARS‐CoV‐2 virus particles. This should provide more physiologic doses (or dose ranges/titration) of spike than the arbitrary doses (3 ug or 5 ug) used in the mouse experiments.

To visualize the acquisition of spike protein and track cells that have acquired the spike protein, we conducted a series of tests and optimizations using different concentrations of Alexa 488 labeled spike protein, ranging from 0.5 to 5 µg. During the processing of lung tissue for microscopic imaging, it was of utmost importance to preserve the integrity of the labeled spike protein in the tissue samples. We determined that instillation of 3 µg of Alexa 488 labeled spike protein yielded the optimal signal strength across the lung sections. Notably, in many mouse models employing intra-nasal instillation protocols for SARS-CoV2 spike protein or RBD domain-only recombinant proteins, a dosage of approximately 3 µg or higher were commonly used. Regarding the titer of spike-incorporated VLPs, it is important to highlight that we did not directly compare the quantity of spike protein present in NL4.3 VLPs to that of the naïve SARS-CoV-2 virus. HIV-1 and SARS-CoV-2 viruses typically carry around 70 gp120 spikes and 30 spikes, respectively.We estimated that SARS-CoV-2 spike-incorporated NL4.3 VLPs may display twice the number of spikes compared to naïve SARS-CoV-2. Notably, our measurements of SARS-CoV-2 spike on NL4.3 VLPs demonstrated similar behavior to SARS-CoV-2 in terms of specific binding to ACE2-expressing 293T cells, indicating their functional similarity in this context.

**Author response image 1. sa3fig1:** Spike protein-incorporated NL4.3 VLPs test with human ACE2-transfected HEK293 cells. The wild-type spike protein-incorporated VLPs and delta envelope NL4.3 VLPs were analyzed using human ACE2-transfected HEK293 cells. The first plot shows ACE2 expression levels in HEK293 cells. The second plot displays the binding pattern of Delta Env NL4.3 VLPs on ACE2-expressing HEK293 cells. The third plot illustrates the binding pattern of wild-type spike protein-incorporated NL4.3 VLPs on ACE2expressing HEK293 cells. The histogram provides a comparison of VLP binding strength to ACE2expressing HEK293 cells.

‐ The PNGase F‐treated protein was not studied in Fig 1. In Fig 2, glycan‐removal by PNGaseF has little effects on cell uptake and cell recruitment in the lung. If binding to one of the Siglec lectins is a critical initial step, experiments should be designed to evaluate this aspect of the spike‐cell interaction in a greater depth.

As the reviewer states results with the PNGase F-treated protein were not shown in Fig. 1 although we showed results in Figs. 2 & 3. See discussion below about our preparation of the PNGase F-treated protein. Perhaps because we elected to use a purified fraction that retained ACE2 binding, the protein we used likely retained some complex glycans. As the reviewer notes the PNGase F treated protein had similar overall cellular recruitment and uptake profiles compared to the untreated spike protein. The PNGase Ftreated fraction we used no longer bound Siglec-F in the flow-based assay, shown in Fig. 7. This argues that the initial uptake and cellular recruitment following intranasal instillation of the Spike protein did not depend upon the engagement of Siglec-F. While Siglec-F on the murine alveolar macrophage can likely efficiently capture the spike proteins other cellular receptors contribute and the overall impact of the spike protein on alveolar macrophages likely reflects its engagement of multiple receptors.

Enzymatic removal of sialic acids from spike may be one parameter to explore. The efficiency of enzymatic removal should also be verified prior to experiments. Finally, the authors need to assess whether the proteins remained functional, folded properly, and did not aggregate.

To obtain the de-glycosylated form of the SARS-CoV-2 spike protein, we employed PNGase F enzymatic digestion to remove glycans. Subsequently, the spike protein was purified using a size exclusion column. During this purification process, the PNGase F-treated spike protein segregated into two distinct fractions, specifically fraction 6 to 8 and fraction 9 to 11 (see revised Figure 1- figure supplement 1).

**Author response image 2. sa3fig2:** Size exclusion chromatography. The peak lines represent the absorbance at 280 nm. PNGase F-treated spike proteins were loaded onto a Superdex 26/60 column, resolved at a flow rate of 1.0 ml/min, and collected in 1 ml fractions.

The Coomassie blue staining of an SDS-PAGE gel revealed that fractions 6 to 8 likely underwent a more pronounced de-glycosylation by PNGase F compared to fractions 9 to 11. Additionally, during the size column purification, we noticed that fraction 6 to 8 exhibited a faster mobility than the untreated spike protein, implying a potentially substantial modification of the protein's conformation. To probe the functional characteristics of the de-glycosylated spike protein in fraction 6 to 8, we conducted binding tests with human ACE2. Strikingly, the spike protein in fraction 6 to 8 completely lost its binding affinity to ACE2, indicating a loss of its ACE2-binding capability. Conversely, the protein in fraction 9 to 11 showed partial de-glycosylation but still retained its original functionality to bind to ACE2 and its antibody.

**Author response image 3. sa3fig3:** FACS analysis of various spike protein-bound beads. Protein bound beads were detected with labeled spike antibody, recombinant human ACE2, and recombinant mouse Siglec-F.

Based on these results, we concluded that fraction 9 to 11 would be the most suitable choice for further studies as the de-glycosylated spike protein, considering its retained functional properties relevant for ligating ACE2 and antibody motifs yet had lost Siglec-F binding. In the revised manuscript we have describe in more detail the purification of the PNGase F treated Trimer and its functional assessment.

‐ Increases in macrophages and alveolar macrophages by Kifunensine Tx spike in Fig 2A suggest effects that are not related to Siglec lectins. These effects are not seen with the wild type or D614 spike trimers, so the relevance of high‐ mannose spike is unclear. On the other hand, there were clear differences between Wuhan and D614 trimers seen in Fig 2A and 2B, but there was no verification to ascertain whether these differences were indeed due to strain differences and not due to batch‐to‐batch variability of the recombinant protein production. The overall glycan contents of the Wuhan and D614 spike protein samples should be measured. If Siglec interaction is the main interest in this study, the terminal sialic acid contents should be determined and compared to those in the corresponding strains in the context of nativeSARS‐CoV‐2 virions.

Our initial observation that Siglec-F positive alveolar macrophages (AMs) avidly acquired spike proteins followed by a rapid leukocyte recruitment provided the rational for us to examine the impact of modifying the glycosylation pattern on the spike protein (de-glycosylated and spike variants) on their binding tropism and their cellular recruitment profiles in the lung. In this context, we examined the influence of several glycan modification on spike proteins, hypothesizing that these modifications would alter the acquisition of the spike protein by mouse AMs compared to the wild-type trimer. While we did not conduct an indepth analysis of the glycan composition and terminal sialic acid contents of the SARS-CoV-2 spike proteins we used we did verify that the different proteins behaved as expected. Most of the biochemical studies were performed in Jim Arthos’ laboratory, which has a long interest in the glycosylation of the HIV envelope protein. On SDS-PAGE the SARS-CoV-2 spike protein purified from the Kifunesine treated CHO cells exhibited a 12 kDa reduction. It bound much better to L-Sign, DC-Sign, and maltose binding lectin, and poorly to Siglec-F. In the cellular studies it bound less well to most of the cellular subsets examined including murine alveolar macrophages. In studies with human blood leukocytes, it relied on cations for binding. However, it retained its toxicity directed at mouse and human neutrophils and it elicited a similar cytokine profile when added to human macrophages. The D614G mutation increased the spike protein binding to P-Selectin, CD163, and snowdrop lectin (mannose binding) suggesting that the mutation had altered the glycan content of the protein. We used the D614G spike protein in a limited number of experiments as it behaved like the wild-type protein except for a slightly altered cellular retention pattern 18 hrs after intranasal instillation. In the revised manuscript we have included its binding to peripheral blood leukocytes. The D614G mutation conferred stronger binding to human monocytes than the original Spike protein. As discussed above, we recovered two fractions following the PNGase F treatment, one with a 40 kDa reduction on SDS-PAGE and the other a 60 kDa decrease and we chose to evaluate the fraction with a 40 kDa reduction in subsequent experiments. Consistent with a loss of N-linked glycans the PNGase F treatment reduced the binding to the lectin PHA, which recognizes complex carbohydrates, and it resulted in a sharp reduction in Siglec-F binding. The lower molecular weight fraction recovered after PNGase F treatment no longer bound ACE2. While our studies showed that alveolar macrophages likely employ Siglec-F as a capturing receptor they possess other receptors that also can capture the spike protein. The downstream consequences of engaging SiglecF and other Siglecs by the SARS-CoV-2 spike protein will require additional studies.

While acknowledging the possibility of some batch-batch variation in recombinant protein preparation, we don’t think this was a major issue. We have noted some batch-batch variations in yield- efficiency, however the purified proteins consistently gave similar results in the various experiments.

‐ Fig 3: The same concern described above applies to the hCoV‐HKU1 spike protein. In Panel D, the PNGase and Kifunensine treatment did not appear to abrogate the neutrophil recruitment. Panel A did not include PNGase and Kif Tx spike proteins. Quantification of images in panel D is missing and should be done on many randomly selected areas.

We analyzed the neutrophil count of images in panel D and the results are presented. (Figure 3-figure supplement 1C). The Kifunensine treatment reduced the neutrophil recruitment at 3 hours, while the PNGase F treated Spike protein recruited as well or slightly more neutrophils. The hCoV-HKU1 S1 domain did not differ much from the saline control.

‐ Fig 4: Kifunensine Tx spike caused more increase in neutrophil damage after intrascrotal injections.PNGase Tx spike was not tested. Connection between Siglec‐spike binding and neutrophil recruitment/damage is lacking.

Exteriorized cremaster muscle imaging functions as a model system for monitoring neutrophil behavior recruited by spike proteins within the local tissue, distinct from Siglec F-positive alveolar macrophages residing in lung tissue. Hence, our primary focus was not on investigating the Siglec/Spike protein interaction. Consequently, we did not utilize PNGase F-treated spike protein in these experiments. To clarify this issue, we added a sentence in main text ‘Although this model lacks Siglec F-positive macrophages, it is worth monitoring the effect of the SARS-CoV-2 Spike protein on neutrophils recruited in the inflammatory local tissue.’

‐ Fig 5. Neutrophil injury was also seen after inhalation (intranasal) of spike protein in mice and in vitro with human neutrophils. Panel B shows no titrating effects of spike (from 0.1 to 2) on Netosis of murine neutrophils. Panel C: Netosis was seen with human neutrophils at 1 but not 0.1. Is this species difference important?

Given the observation of neutrophil NETosis in the mouse imaging experiment, our objective was to characterize the direct impact of the spike protein on human and murine neutrophils. The origins of the neutrophils are different as the murine neutrophils were purified from mouse bone marrow while the human neutrophils were purified from human blood. Both purification protocols led to greater than 98% neutrophils. However, the murine neutrophils contain many more immature cells (50-60%) because the bone marrow served as their source. Furthermore, the murine neutrophils are from 6–8-week-old mice while the human neutrophils are from 30-50 year-old humans. More work would be needed to sort out whether there is any difference between human and mouse neutrophils in their propensity to undergo netosis in response to Spike protein.

‐ Kifunensine Tx again did not cause any reduction, indicating the lack of involvement of sialic acid. How was this related to Siglec participation directly or indirectly? There was no quantification for Panel D.

We do not think that Siglecs play a role in the induction of neutrophil netosis as the Spike proteins lacking Siglec interactions induced similar levels of netosis. Likely other neutrophil receptors are important. As noted in the text,

"human neutrophils express several C-type lectin receptors including CLEC5A, which has been implicated in SARS-CoV-2 triggered neutrophil NETosis." Our goal with the data in Panel D was to visualize human neutrophil NETosis on trimer-bearing A549 cells we relied on the flow cytometry assays for quantification.

‐ The rationale for testing cation dependence is unclear and should be described. What is the significance of "cations enhanced leukocyte binding particularly so with the high mannose protein"? Are there cationdependent receptors for spike independent of glycans and huACE‐2? If so, how is this relevant to the main topic of this paper?

It is well known that many glycan bindings by C-type lectins are calcium-dependent, involving specific amino acid residues that coordinate with calcium ions and bind to the hydroxyl groups of sugars. As discussed in our previous draft, the C-type lectin receptor L-SIGN has been suggested as a calciumdependent receptor for SARS-CoV-2, specifically interacting with high-mannose-type N-glycans on the SARS-CoV-2 spike protein. Therefore, it was worthwhile to investigate the calcium-dependent manner of spike protein binding to various types of immune cells. We added some data to this figure. It now includes the binding profile of the D614G protein. In addition, we corrected the binding data by subtracting the fluorescent signal from the unstained control cells.

‐ Fig 7: human Siglec 5 and 8 were studied in comparison with mouse Siglec F. Recombinant protein data are not congruent with transfected 293 cell data. Panel A, the best binding to hSiglec 5 and 8 are the PNGase F Tx spike protein; how to interpret these data? Panel B: only the WT and D614G spike proteins binding to Siglec 5 and 8 on transfected cells. It made sense that kif Tx (high‐mannose) and PNGaseF Tx(no glycan) spike would not bind to the Siglecs, but they did not bind to ACE2 either, indicative of nonfunctional spike proteins.

We discussed this as follows: ‘The closest human paralog of mouse Siglec-F is hSiglec-8 (reference 40). While expressed on human eosinophils and mast cells, human AMs apparently lack it. In contrast, human AMs do express Siglec-5 (reference 37). Along with its paired receptor, hSiglec-14, Siglec-5 can modulate innate immune responses (reference 41). When tested in a bead binding assay, in contrast to Siglec-F, neither hSiglec-5 or -8 bound the recombinant spike protein, yet their expression in a cellular context allowed binding. The in vitro bead binding assay we established demonstrated the specific binding of the bait molecule to target molecules. However, it does have limitations in replicating the complexities of the actual cellular environment. As discussed previously the PNGase Tx fraction we used in these experiments retained ACE2 binding, but loss binding to Siglec-F in the bead assay. In a biacore assay, not shown, the PNGase Tx fraction bound L-Sign and DC-Sign better than the untreated trimer, and it retained human ACE2 binding although it bound less well than wild type-trimer. Why the PNGase Tx fractions bound poorly to the human ACE2 transfected HEK293 cells is unclear. A higher density of recombinant ACE2 on the beads compared to that expressed on the surface of HEK293 may explain the difference. Alternatively in the bead assay we used a recombinant human ACE2-Fc fragment fusion protein purified from HEK293 cells, while in the transfection assay, we expressed human full length ACE2. The biacore, the bead binding, and the functional assays we performed all suggest that we had used intact recombinant proteins.

‐ Fig 8: This last set of experiment was to measure cytokine release by different types of macrophage cultures treated with spike from different cells with vs without Kifunensine Tx. The connection of these experiments to the rest is tenuous and is not explained. This is one of the examples where bits of data are presented without tying them together.

Dysregulated cytokine production significantly contributes to the pathogenesis of severe COVID-19 infection. Since we had observed strong binding of the spike protein to human monocytes and murine alveolar macrophages, we tested whether the spike protein altered cytokine production by human monocyte-derived macrophages. Depending on the culture conditions human monocytes can be differentiated M0, M1, or M2 phenotypes. Each type of macrophage responds differently to stimulants, often leading to distinct patterns of cytokine secretion. These patterns offer valuable insights into the immune response. The cytokine profiling conducted in this study enhances our understanding of how distinct macrophage types react to the spike protein.

‐ Discussion section did not describe how the various experiments and data are tied together. The authors explained the interactions of spike with different cell types in each paragraph separately, leaving this reviewer really confused as to what the authors want to convey as the main message of the paper.

We have modified discussion to address this issue.

**Reviewer #3 (Recommendations For The Authors):**
‐ The authors may want to refer to "intranasal instillation" to distinguish it from inhalation of an aerosolised liquid. How was the dose of the spike protein selected? There is some dose information in different settings, but usually between 0.1‐1 µg/ml or 0.1 µg‐5 µg range for in vivo injection, but the rationale for these ranges should be discussed. Is this mimicking a real situation during infections or a condition that might be used for vaccines?

While inhalation of aerosolized liquid closely mimics the natural route of human exposure to respiratory infectious materials, intranasal instillation with a liquid inoculum remains a widely accepted standard approach for virus or vaccine inoculation across various laboratory species. To clearly define our mouse model, we are changing the term 'inhalation' to 'instillation'. We previously answered to Reviewer #2 as following: To visualize the acquisition of spike protein and track cells that have acquired the spike protein, we conducted a series of tests and optimizations using different concentrations of Alexa Fluor 488 labeled spike protein, ranging from 0.5 to 5 µg. During the processing of lung tissue for microscopic imaging, it was of utmost importance to preserve the integrity of the labeled spike protein on the tissue samples. Through our investigations, we determined that an instillation of 3 µg of Alexa Fluor 488 labeled spike protein yielded the most optimal signal strength across the lung sections. Notably, in many mouse models employing intra-nasal instillation protocols for SARS-CoV-2 spike protein or RBD domain-only recombinant proteins, a dosage of approximately 3 µg or higher was commonly used. Hence, based on these references and our preliminary studies, we selected 3 µg as the optimal concentration of instilled spike protein per mouse.

‐ Controls are not evenly applied. In some cases, the control for the large and complex SARS‐CoV2 spiker trimer is PBS. This seems insufficient to control against effects of injecting such complex proteins that can undergo significant conformational changes after uptake by a cell. In some cases, human coronavirus spike proteins from different viruses are used, but not much is said about these proteins and the different glycoforms are not explored. Are these prepared in the same way and do they have similar glycoforms.For example, if the Siglecs bind sialic acid on N‐linked glycans, then why do the purified Siglecs or Siglecs expressed in cells not bind the HKU‐1 spike, which would have such sialic acids if expressed in the same way as the CoV2 spike?

We have taken careful consideration to select an appropriate control material for these experiments. Initially, we opted to employ Saline or PBS for intranasal instillation as a vehicle control, a choice aligned with the approach taken in numerous previous studies involving lung inflammation mouse models. However, as the reviewer pointed out, we share the concern for achieving more meaningful and comparable control materials, particularly considering the size and complexity of the recombinant protein. In accordance with this perspective, we introduced glycan-modified spike proteins and the HCoV-HKU1 S1 subunit. Figure 3 illustrates our comprehensive evaluation of various spike proteins in terms of their impact on neutrophil recruitment. The diversity of sialic acid structures observed on recombinant proteins expressed within the same cell emerges from the intricate interplay of multiple factors within the cellular glycosylation machinery. This complex enzymatic process empowers cells to finely modulate glycan structures and sialic acid patterns, tailoring them to suit the diverse biological functions of distinct proteins. Despite structural similarities between the HCoV-HKU1 and SARS-CoV-2 spike proteins, their glycan modifications vary, thereby leading to distinct binding properties with various Siglec subtypes. All recombinant proteins used in this study except for the S1 subunits were generated within our laboratory. These include the wild-type spike protein, the D614G Spike protein, the Kifunensine-treated high mannose spike proteins, and the PNGase F-treated deglycosylated spike proteins. All the proteins were produced using the same protocol using CHO cells or on occasion HEK293F cells. We have indicated in the manuscript where we used HEK293F cells for the protein production otherwise they were produced in CHO cells.

‐ Figure 1 F‐I, there should be a control for VLP without SARS‐CoV2 spike as the VLP will contain other components that may be active in the system.

We tested the delta Env VLP for alveolar macrophage acquisition and neutrophil recruitment. We found a similar alveolar macrophage acquisition of the VLPs, but significantly less neutrophil recruitment compared to the free Spike protein. Since the uptake pattern with the VLPs matched that of the spike protein we did not consider adding a non-spike bearing VLP as a control. The rapid VLPs clearance into the lymphatics shortly after instillation may account for the reduced neutrophil recruitment following their instillation (Figure 1 figure supplement 2B, C).

‐ In Figure 1H, that do they mean by autofluorescence? Is this the cyan signal?Is the green signal also autofluorescence as this is identified as the VLP?

We appreciate reviewer pointing out the typo regarding autofluorescence in the figure image. To provide clarity regarding the background in all lung section images, we have included additional supplemental data. During the fixation process of lung tissue, various endogenous elements in the tissue sample contribute to autofluorescence when exposed to lasers in the confocal microscope. Specifically, collagen and elastin present in the lung vasculature, including airways and blood vessels, are dominant structures that generate autofluorescence. To address this issue, we have implemented optimizations to distinguish between real signals and the noise caused by autofluorescence. We inadvertently failed to indicate the source of the strong cyan signal. The signal is due to Evans Blue dye delineating lung airway structures, which contain collagen and elastin—known binding materials for Evans Blue dye. This explains the strong fluorescence signals observed in the airways. We conjugated the recombinant spike protein with Alexa Fluor 488, and viral-like particles (VLPs) were visualized with gag-GFP. (Figure 1 figure supplement 2A, D)

‐ The control for SARS‐CoV2 spike trimer is PBS, but how can the authors distinguish patterns specific to the spike trimer from any other protein delivered by intranasal instillation. Could they use another channel with a control glycoprotein to determine if there is anything unique about the pattern for spike trimer?

Alveolar macrophages employ numerous receptors to capture glycoproteins that have mannose, Nacetylglucosamine, or glucose exposed. Galactose-terminal glycoproteins are typically not bound. We do not think that the Spike protein is unique in its propensity to target alveolar macrophages.

‐ What is the parameter measured in Figure S2B?

The percentage of the different cell types that have retained the instilled Spike protein at the three-hour time point. .

‐ The Spike trimer with high mannose oligosaccharides may gain binding to the mannose receptor. It may be helpful to state the distribution of this receptor and comment is it could be responsible for this having the largest effect size for some cell types.

We agree that the spike trimer with high mannose should target cells bearing the mannose receptor. We have modified the discussion to address this point and have mentioned some of the cell types likely to bind the high mannose bearing spike protein.

‐ A key experiment is the Evans Blue measure of lung injury in Figure 3A. A control with the HKU‐1 spike is also performed, but more details on the matching of this proteins production to the SARS‐CoV2 spike trimer and the quantification of these comparative result should be provided. To show that the SARSCoV2 spike trimer can cause tissue injury on its own seems like a very important result, but the impact is currently reduced by the inconsistent application of controls and quantification of key results.Furthermore, if these results can be repeated in the B6 and B6 K18‐hACE2 mouse model it might further increase the impact by demonstrating whether or not hACE2 contributes to this effect.

We repeated the lung permeability assay using the S1 subunit from the original SARS-CoV-2 and the S1 subunit from HCoV-HKU1. Both proteins were made by the same company using a similar expression system and purification protocol. Consistent with our original data, the instillation of the SARS-CoV-2 S1 subunit led to an increase in lung vasculature permeability, whereas the HCoV-HKU-1 S1 subunit had a minimal impact. (Figure 3 figure supplement 1A). This experiment suggests that it the S1 subunit that leads to the increase in vascular permeability. To address the contribution of hACE2 in this phenomenon, we conducted a lung permeability assay using K18-hACE2 transgenic mice. The K18-hACE2 transgenic mice exhibited a slight increase in lung vasculature permeability upon SARS-CoV-2 trimer instillation compared to the non-transgenic mice. This suggests that the hACE2-Spike protein interaction may contribute to an increase in lung vascular permeability during SARS-CoV-2 lung infection (Figure 3 figure supplement 1B).

‐ For Figure 4A, could they provide quantification. The neutrophil extravasation with Trimer appears quite robust, but the authors seem to down‐play this and it's not clear without quantification.

To address this issue, we analyzed and graphed the neutrophil numbers in each image. Injection of the trimer along with IL-1β significantly increased neutrophil infiltration. (Figure 4 figure supplement 1)

‐ In Figure 4B, there are no neutrophils at all in the BSA condition. Is this correct? Intravascular neutrophils were detected with PBS injection in Figure 4A.

We demonstrated that the neutrophil behaviors occur within the infiltrated tissue rather than within the blood vessels. Even when examining the blood vessels in all other images, it is challenging to identify neutrophils adhering to the endothelium of the blood vessels. Neutrophils observed in the PBS 3-hour control group are likely acute responders to the local injection, as a smaller number of neutrophils were observed in the 6-hour image.

‐ In Figure 5A the observation of neutrophil response in lung slices seems to be presented an anecdotal account. The neutrophil appears to polarize, but is this a consistent observation? How many such observations were made?

We have consistent observations across three different experiments. In addition, highly polarized and fragmented neutrophils were consistently observed in the fixed lung section images.

‐ The statement: "human Siglec‐5 and Siglec‐8 bound poorly despite being the structural and functional equivalents of Siglec F, respectively (37)". How can one Siglec be the structural and the other the functional equivalent of Siglec‐F? It might help to provide a little more detail as to how these should be seen.

Mouse Siglec-F has two distinct counterparts in the human Siglec system, both in terms of structure and function. In the context of domain structure, human Siglec-5 serves as the counterpart to mouse Siglec-F. However, it's important to note that while human Siglec-8 is not a genetic ortholog of mouse Siglec-F, it is expressed on similar cellular populations and functions as a functional paralog.

‐ The assay using purified proteins and proteins expressed in cells don't fully agree. For example, it's very surprising that recombinant Siglec 5 and 8 bind better to the non‐glycosylated form than to the glycosylated trimer. It appears from Figure S1 that the PNGaseF treated Spike contains at least partly glycosylated monomers and it also appears that the Kifunesine effect may be partial. PNGaseF may have a hard time removing some glycans from a native protein.

We were also surprised by the results using the PNGase F treated Spike protein in that it lost binding to Siglec-F and retained binding to human Siglec-5 and 8 in the bead assay, shown in Figure 7A. As explained above we used a purified fraction of the PNGase F treated protein that retained some functional activity as assessed in the ACE2 binding assay and in biacore assays not shown. The persistent binding of Siglec-5 and Siglec-8 suggests that removal of some of the complex glycans had revealed sites capable of binding Siglec-5 and 8. We would agree with the reviewer that the PNGase treatment we used only removed some of the glycans from the native protein. In data not shown the high mannose spike protein behaved as predicted in biacore assays, binding better to DC-SIGN and maltose binding lectin, but less well to PHA and less well to ACE2. The high mannose trimer also bound less to the HEK293 cells expressing ACE2, Siglec-5, or Siglec-8 as well as peripheral blood leukocytes.